# Pessimistic Nonlinear Least-Squares Value Iteration for Offline Reinforcement Learning

**Qiwei Di**[1], **Heyang Zhao**[1], **Jiafan he**[1], **Quanquan Gu**[1]
[1]Department of Computer Science, University of California, Los Angeles
`{qiwei2000,hyzhao,jiafanhe19,qgu}@cs.ucla.edu,`

## Abstract

Offline reinforcement learning (RL), where the agent aims to learn the optimal policy based on the data collected by a behavior policy, has attracted increasing attention in recent years. While offline RL with linear function approximation has been extensively studied with optimal results achieved under certain assumptions, many works shift their interest to offline RL with non-linear function approximation. However, limited works on offline RL with non-linear function approximation have instance-dependent regret guarantees. In this paper, we propose an oracle-efficient algorithm, dubbed Pessimistic Nonlinear Least-Square Value Iteration (PNLSVI), for offline RL with non-linear function approximation. Our algorithmic design comprises three innovative components: (1) a variance-based weighted regression scheme that can be applied to a wide range of function classes, (2) a subroutine for variance estimation, and (3) a planning phase that utilizes a pessimistic value iteration approach. Our algorithm enjoys a regret bound that has a tight dependency on the function class complexity and achieves minimax optimal instance-dependent regret when specialized to linear function approximation. Our work extends the previous instance-dependent results within simpler function classes, such as linear and differentiable function to a more general framework.

## 1 Introduction

Offline reinforcement learning (RL), also known as batch RL, is a learning paradigm where an agent learns to make decisions based on a set of pre-collected data, instead of interacting with the environment in real-time like online RL. The goal of offline RL is to learn a policy that performs well in a given task, based on historical data that was collected from an unknown environment. Recent years have witnessed significant progress in developing offline RL algorithms that can leverage large amounts of data to learn effective policies. These algorithms often incorporate powerful function approximation techniques, such as deep neural networks, to generalize across large state-action spaces. They have achieved excellent performances in a wide range of domains, including the games of Go and chess (Silver et al., 2017; Schrittwieser et al., 2020), robotics (Gu et al., 2017; Levine et al., 2018), and control systems (Degrave et al., 2022).

Several works have studied the theoretical guarantees of offline tabular RL and proved near-optimal sample complexities in this setting (Xie et al., 2021b; Shi et al., 2022; Li et al., 2022). However, these algorithms cannot handle real-world applications with large state and action spaces. Consequently, a significant body of research has devoted to offline RL with function approximation. For example, (Jin et al., 2021b) proposed the first efficient algorithm for offline RL with linear MDPs, employing the principle of pessimism. Subsequently, numerous works have presented a range of algorithms for offline RL with linear function approximation, as seen in Zanette et al. (2021); Min et al. (2021); Yin et al. (2022a); Xiong et al. (2023); Nguyen-Tang et al. (2023). Among them, some works have instance-dependent (a.k.a., problem-dependent) upper bound (Jin et al., 2021b; Yin et al., 2022a; Xiong et al., 2023; Nguyen-Tang et al., 2023), which matches the worst-case result when dealing with the "hard instance" and performs better in easy cases.

To address the complexities of working with more complex function classes, recent research has shifted the focus towards offline reinforcement learning (RL) with general function approximation (Chen & Jiang, 2019; Xie et al., 2021a). Utilizing the principle of pessimism first used in Jin et al. (2021b), Xie et al. (2021a) enforced pessimism at the initial state over the set of functions consistent with the Bellman equations. Their algorithm requires solving an optimization problem over all the potential policies and corresponding version space, which includes all functions with lower Bellman-error. To overcome this limitation, Xie et al. (2021a) proposed a practical algorithm which

has a poor dependency on the function class complexity. Later, Cheng et al. (2022) proposed an adversarially trained actor critic method based upon the concept of relative pessimism. Their result is not statistically optimal and was later improved by Zhu et al. (2023). Both algorithms' implementation relies on a no-regret policy optimization oracle. Another line of works, such as (Zhan et al., 2022; Ozdaglar et al., 2023; Rashidinejad et al., 2021), sought to solve the offline RL problem through a linear programming formulation, which requires some additional convexity assumptions on the policy class. However, most of these works only have worst-case regret guarantee. The only exception is Yin et al. (2022b), which studies the general differentiable function class and proposed an LSVI-type algorithm. For more general function classes, how to get instance-dependent characterizations is still an open problem. Therefore, a natural question arises:

*Can we design a computationally tractable algorithm that is statistically efficient with respect to the complexity of nonlinear function class and has an instance-dependent regret bound?*

We give an affirmative answer to the above question in this work. Our contributions are listed as follows:

- We propose a pessimism-based algorithm Pessimistic Nonlinear Least-Square Value Iteration (PNLSVI) designed for nonlinear function approximation, which strictly generalizes the existing pessimism-based algorithms for both linear and differentiable function approximation (Xiong et al., 2023; Yin et al., 2022b). Our algorithm is oracle-efficient, i.e., it is computationally efficient when there exists an efficient regression oracle and bonus oracle for the function class (e.g., generalized linear function class). The bonus oracle can also be reduced to a finite number of calls to the regression oracle.

- We introduce a new type of $D^2$-divergence to quantify the uncertainty of an offline dataset, which naturally extends the role of the elliptical norm seen in the linear setting and the $D^2$-divergence in Gentile et al. (2022); Agarwal et al. (2023); Ye et al. (2023) for online RL. We prove an instance-dependent regret bound characterized by this new $D^2$-divergence. Our regret bound has a tight dependence on complexity of the function class, i.e., $\widetilde{O}(\sqrt{\log \mathcal{N}})$ with $\mathcal{N}$ being the cardinality of the underlying function class, which improves the $\widetilde{O}(\log \mathcal{N})$ dependence [1] in (Yin et al., 2022b) and resolves the open problem raised in their paper.

**Notation:** In this work, we use lowercase letters to denote scalars and use lower and uppercase boldface letters to denote vectors and matrices respectively. For a vector $\mathbf{x} \in \mathbb{R}^d$ and matrix $\mathbf{\Sigma} \in \mathbb{R}^{d \times d}$, we denote by $\|\mathbf{x}\|_2$ the Euclidean norm and $\|\mathbf{x}\|_{\mathbf{\Sigma}} = \sqrt{\mathbf{x}^\top \mathbf{\Sigma} \mathbf{x}}$. For two sequences $\{a_n\}$ and $\{b_n\}$, we write $a_n = O(b_n)$ if there exists an absolute constant $C$ such that $a_n \leq C b_n$, and we write $a_n = \Omega(b_n)$ if there exists an absolute constant $C$ such that $a_n \geq C b_n$. We use $\widetilde{O}(\cdot)$ and $\widetilde{\Omega}(\cdot)$ to further hide the logarithmic factors. For any $a \leq b \in \mathbb{R}$, $x \in \mathbb{R}$, let $[x]_{[a,b]}$ denote the truncate function $a \cdot \mathbb{1}(x \leq a) + x \cdot \mathbb{1}(a \leq x \leq b) + b \cdot \mathbb{1}(b \leq x)$, where $\mathbb{1}(\cdot)$ is the indicator function. For a positive integer $n$, we use $[n] = \{1, 2, .., n\}$ to denote the set of integers from 1 to $n$.

## 2 RELATED WORK

**RL with function approximation.** As one of the simplest function approximation classes, linear representation in RL has been extensively studied in recent years (Jiang et al., 2017; Dann et al., 2018; Yang & Wang, 2019; Jin et al., 2020; Wang et al., 2020c; Du et al., 2019; Sun et al., 2019; Zanette et al., 2020a;b; Weisz et al., 2021; Yang & Wang, 2020; Modi et al., 2020; Ayoub et al., 2020; Zhou et al., 2021; He et al., 2021; Zhong et al., 2022). Several assumptions on the linear structure of the underlying MDPs have been made in these works, ranging from the *linear MDP* assumption (Yang & Wang, 2019; Jin et al., 2020; Hu et al., 2022; He et al., 2022; Agarwal et al., 2023) to the *low Bellman-rank* assumption (Jiang et al., 2017) and the *low inherent Bellman error* assumption (Zanette et al., 2020b). Extending the previous theoretical guarantees to more general problem classes, RL with nonlinear function classes has garnered increased attention in recent years (Wang et al., 2020b; Jin et al., 2021a; Foster et al., 2021; Du et al., 2021; Agarwal & Zhang, 2022; Agarwal et al., 2023). Various complexity measures of function classes have been studied including Bellman rank (Jiang et al., 2017), Bellman-Eluder dimension (Jin et al., 2021a), Decision-Estimation Coefficient (Foster et al., 2021) and generalized Eluder dimension (Agarwal et al., 2023). Among these works, the setting in our paper is most related to Agarwal et al. (2023) where $D^2$-divergence (Gentile et al., 2022) was introduced in RL to indicate the uncertainty of a sample with respect to a particular sample batch.

---

[1] In (Yin et al., 2022b), they denote by $d$ the complexity of the underlying function class, which is essentially $\log \mathcal{N}$ using our notation.

**Offline tabular RL.** There is a line of works integrating the principle of pessimism to develop statistically efficient algorithms for offline tabular RL setting (Rashidinejad et al., 2021; Yin & Wang, 2021; Xie et al., 2021b; Shi et al., 2022; Li et al., 2022). More specifically, Xie et al. (2021b) utilized the variance of transition noise and proposed a nearly optimal algorithm based on pessimism and Bernstein-type bonus. Subsequently, Li et al. (2022) proposed a model-based approach that achieves minimax-optimal sample complexity without burn-in cost for tabular MDPs. Shi et al. (2022) also contributed by proposing the first nearly minimax-optimal model-free offline RL algorithm.

**Offline RL with linear function approximation.** Jin et al. (2021b) presented the initial theoretical results on offline linear MDPs. They introduced a pessimism-principled algorithmic framework for offline RL and proposed an algorithm based on LSVI (Jin et al., 2020). Min et al. (2021) subsequently considered offline policy evaluation (OPE) in linear MDPs, assuming independence between data samples across time steps to obtain tighter confidence sets and proposed an algorithm with optimal $d$ dependence. Yin et al. (2022a) took one step further and considered the policy optimization in linear MDPs, which implicitly requires the same independence assumption. Zanette et al. (2021) proposed an actor-critic-based algorithm that establishes pessimism principle by directly perturbing the parameter vectors in a linear function approximation framework. Recently, Xiong et al. (2023) proposed a novel uncertainty decomposition technique via a reference function, and demonstrated their algorithm matches the performance lower bound up to logarithmic factors.

**Offline RL with general function approximation.** Chen & Jiang (2019) examined the assumptions underlying value-function approximation methods and established an information-theoretic lower bound. Xie et al. (2021a) introduced the concept of Bellman-consistent pessimism, which enables sample-efficient guarantees by relying solely on the Bellman-completeness assumption. Uehara & Sun (2021) focused on model-based offline RL with function approximation under partial coverage, demonstrating that realizability in the function class and partial coverage are sufficient for policy learning. Zhan et al. (2022) proposed an algorithm that achieves polynomial sample complexity under the realizability and single-policy concentrability assumptions. Nguyen-Tang & Arora (2023) proposed a method of random perturbations and pessimism for neural function approximation. For differentiable function classes, Yin et al. (2022b) made advancements by improving the sample complexity with respect to the planing horizon $H$. However, their result had an additional dependence on the dimension $d$ of the parameter space, whereas in linear function approximation, the dependence is typically on $\sqrt{d}$. Recently, a sequence of works focus on proposing statistically optimal and practical algorithms under single concentrability assumption. Ozdaglar et al. (2023) provided a new reformulation of linear-programming. Rashidinejad et al. (2022) used the augmented Lagrangian method and proved augmented Lagrangian is enough for statistically optimal offline RL. Cheng et al. (2022) proposed an actor-critic algorithm by formulating the offline RL problem into a Stackelberg game. However, their result is not statistically optimal and Zhu et al. (2023) improved it by combining the marginalized importance sampling framework and achieved the optimal statistical rate. We leave a comparison of these works in Appendix A.

## 3 PRELIMINARIES

In our work, we consider the inhomogeneous episodic Markov Decision Processes (MDP), which can be denoted by a tuple of $\mathcal{M}(\mathcal{S}, \mathcal{A}, H, \{r_h\}_{h=1}^H, \{\mathbb{P}_h\}_{h=1}^H)$. In specific, $\mathcal{S}$ is the state space, $\mathcal{A}$ is the finite action space, $H$ is the length of each episode. For each stage $h \in [H]$, $r_h : \mathcal{S} \times \mathcal{A} \to [0, 1]$ is the reward function[2] and $\mathbb{P}_h(s'|s, a)$ is the transition probability function, which denotes the probability for state $s$ to transfer to next state $s'$ with current action $a$. A policy $\pi := \{\pi_h\}_{h=1}^H$ is a collection of mappings $\pi_h$ from a state $s \in \mathcal{S}$ to the simplex of action space $\mathcal{A}$. For simplicity, we denote the state-action pair as $z := (s, a)$. For any policy $\pi$ and stage $h \in [H]$, we define the value function $V_h^\pi(s)$ and the action-value function $Q_h^\pi(s, a)$ as the expected cumulative rewards starting at stage $h$, which can be denoted as follows:

$$Q_h^\pi(s, a) = r_h(s, a) + \mathbb{E}\bigg[\sum_{h'=h+1}^H r_{h'}\big(s_{h'}, \pi_{h'}(s_{h'})\big)\big|s_h = s, a_h = a\bigg], \ V_h^\pi(s) = Q_h^\pi\big(s, \pi_h(s)\big),$$

where $s_{h'+1} \sim \mathbb{P}_h(\cdot|s_{h'}, a_{h'})$ denotes the observed state at stage $h' + 1$. By this definition, the value function $V_h^\pi(s)$ and action-value function $Q_h^\pi(s, a)$ are bounded in $[0, H]$. In addition, we define the optimal value function $V_h^*$ and the optimal action-value function $Q_h^*$ as

---

[2]While we study the deterministic reward functions for simplicity, it is not difficult to generalize our results to stochastic reward functions.

$V_h^*(s) = \max_\pi V_h^\pi(s)$ and $Q_h^*(s,a) = \max_\pi Q_h^\pi(s,a)$. We denote the corresponding optimal policy by $\pi^*$. For any function $V : \mathcal{S} \to \mathbb{R}$, we denote $[\mathbb{P}_h V](s,a) = \mathbb{E}_{s' \sim \mathbb{P}_h(\cdot|s,a)} V(s')$ and $[\text{Var}_h V](s,a) = [\mathbb{P}_h V^2](s,a) - \left([\mathbb{P}_h V](s,a)\right)^2$ for simplicity. For any function $f : \mathcal{S} \times \mathcal{A} \to \mathbb{R}$, we define $f(s) = \max_a f(s,a)$. For any function $f : \mathcal{S} \to \mathbb{R}^3$, we define the Bellman operator $\mathcal{T}_h$ as $[\mathcal{T}_h f](s_h, a_h) = \mathbb{E}_{s_{h+1} \sim \mathbb{P}_h(\cdot|s_h, a_h)} [r_h(s_h, a_h) + f(s_{h+1})]$. Based on this definition, for every stage $h \in [H]$ and policy $\pi$, we have the following Bellman equation for value functions $Q_h^\pi(s,a)$ and $V_h^\pi(s)$, as well as the Bellman optimality equation for optimal value functions:

$$Q_h^\pi(s_h, a_h) = [\mathcal{T}_h V_{h+1}^\pi](s_h, a_h), \ Q_h^*(s_h, a_h) = [\mathcal{T}_h V_{h+1}^*](s_h, a_h),$$

where $V_{H+1}^\pi(s) = V_{H+1}^*(s) = 0$. We also define the Bellman operator for second moment as $[\mathcal{T}_{2,h} f](s_h, a_h) = \mathbb{E}_{s_{h+1} \sim \mathbb{P}_h(\cdot|s_h, a_h)} \left[\left(r_h(s_h, a_h) + f(s_{h+1})\right)^2\right]$. For simplicity, we omit the subscripts $h$ in the Bellman operator without causing confusion.

**Offline Reinforcement Learning:** In offline RL, the agent only has access to a batch-dataset $D = \{s_h^k, a_h^k, r_h^k : h \in [H], k \in [K]\}$, which is collected by a behavior policy $\mu$, and the agent cannot interact with the environment. We also make the compliance assumption of the dataset:

**Assumption 3.1.** For a dataset $\mathcal{D} = \{(s_h^k, a_h^k, r_h^k)\}_{k,h=1}^{K,H}$, let $\mathbb{P}_\mathcal{D}$ be the joint distribution of the data collecting process. We say $\mathcal{D}$ is compliant with an underlying MDP $(\mathcal{S}, \mathcal{A}, H, \mathbb{P}, r)$ if

$$\mathbb{P}_\mathcal{D}(r_h^k = r', s_{h+1}^k = s' \mid \{(s_h^j, a_h^j)\}_{j=1}^k, \{(r_h^j, s_{h+1}^j)\}_{j=1}^{k-1})$$
$$= \mathbb{P}_h(r_h(s_h, a_h) = r', s_{h+1} = s' \mid s_h = s_h^k, a_h = a_h^k)$$

for all $r' \in [0,1], s' \in \mathcal{S}, h \in [H]$ and $k \in [K]$.

This assumption is common in offline RL and has also been made in Jin et al. (2021b); Zhong et al. (2022). When the dataset originates from one behavior policy, this assumption naturally holds. In this paper, we assume our dataset is generated by a single behavior policy $\mu$. In addition, for each stage $h$, we denote the induced distribution of the state-action pair by $d_h^\mu$.

Given the batch dataset, the goal of offline RL is finding a near-optimal policy $\pi$ that minimizes the suboptimality $V_1^*(s) - V_1^\pi(s)$. For simplicity, and when there's no risk of confusion, we will use the shorthand notation $z_h^k = (s_h^k, a_h^k)$ to denote the state-action pair in the dataset up to stage $h$ and episode $k$.

**General Function Approximation:** Given a general function class $\{\mathcal{F}_h\}_{h \in [H]}$, where each function class $\mathcal{F}_h$ is composed of functions $f_h : \mathcal{S} \times \mathcal{A} \to [0, L]$. For simplicity, we assume $L = O(H)$ throughout the paper. We make the following assumptions on the function class.

**Assumption 3.2** ($\epsilon$-realizability under general function approximation). For each stage $h \in [H]$, there exists a function $f_h^* \in \mathcal{F}_h$ close to the optimal value function such that $\|f_h^* - Q_h^*\|_\infty \leq \epsilon$.

**Assumption 3.3** ($\epsilon$-completeness under general function approximation, Agarwal et al. 2023). We assume for each stage $h \in [H]$, and any function $V : \mathcal{S} \to [0, H]$, there exists functions $f_h, f_{2,h} \in \mathcal{F}_h$ such that

$$\max_{(s,a) \in \mathcal{S} \times \mathcal{A}} |f_h(s,a) - [\mathcal{T}_h V](s,a)| \leq \epsilon, \text{ and } \max_{(s,a) \in \mathcal{S} \times \mathcal{A}} |f_{2,h}(s,a) - [\mathcal{T}_{2,h} V](s,a)| \leq \epsilon.$$

In this paper, for simplicity, we assume that the function class is finite and denote its cardinality by $\mathcal{N} = \max_{h \in [H]} |\mathcal{F}_h|$. For infinite function classes, we can use the covering number to replace the cardinality. Note that the covering number will be reduced to the cardinality when the function class is finite.

We introduce the following definition ($D^2$-divergence) to quantify the disparity of a given point $z = (s,a)$ from the historical dataset $\mathcal{D}_h$. It is a reflection of the uncertainty of the dataset.

**Definition 3.4.** For a function class $\mathcal{F}_h$ consisting of functions $f_h : \mathcal{S} \times \mathcal{A} \to \mathbb{R}$ and $\mathcal{D}_h = \{(s_h^k, a_h^k, r_h^k)\}_{k \in [K]}$ as a dataset that corresponds to the observations collected up to stage $h$ in the MDP, we introduce the following $D^2$-divergence:

$$D_{\mathcal{F}_h}^2(z; \mathcal{D}_h; \sigma_h^2) = \sup_{f_1, f_2 \in \mathcal{F}_h} \frac{(f_1(z) - f_2(z))^2}{\sum_{k \in [K]} \frac{1}{(\sigma_h(z_h^k))^2} (f_1(z_h^k) - f_2(z_h^k))^2 + \lambda},$$

where $\sigma_h^2(\cdot, \cdot) : \mathcal{S} \times \mathcal{A} \to \mathbb{R}$ is a weight function.

---

[3] In this paper, we slightly abuse notation $f$ for both $f(s)$ and $f(s,a)$. The context readily clarifies the intended meaning, i.e., value function or action-value function.

**Remark 3.5.** This definition signifies the extent to which the behavior of functions within the function class can deviate at the point $z = (s, a)$, based on their difference in the historical dataset. It can be viewed as the generalization of the weighted elliptical norm $\|\phi(s, a)\|_{\Sigma_h^{-1}}$ in linear case, where $\phi$ is the feature map and $\Sigma_h$ is defined as $\sum_{k \in [K]} \sigma_h^{-2}(s_h^k, a_h^k) \phi(s_h^k, a_h^k) \phi(s_h^k, a_h^k)^\top + \lambda \mathbf{I}$. Similar ideas have been used to define the Generalized Eluder dimension for online RL (Gentile et al., 2022; Agarwal et al., 2023; Ye et al., 2023). In these works, the summation is over a sequence up to the $k$-th episode rather than the entire historical dataset.

**Data Coverage Assumption:** In offline RL, there exists a discrepancy between the state-action distribution generated by the behavior policy and the distribution from the learned policy. Under this situation, the distribution shift problem can cause the learned policy to perform poorly or even fail in offline RL. In this work, we consider the following data coverage assumption to control the distribution shift.

**Assumption 3.6** (Uniform Data Coverage). there exists a constant $\kappa > 0$, such that for any stage $h$ and functions $f_1, f_2 \in \mathcal{F}_h$, the following inequality holds,

$$\mathbb{E}_{d_h^\mu} \left[ \left( f_1(s_h, a_h) - f_2(s_h, a_h) \right)^2 \right] \geq \kappa \|f_1 - f_2\|_\infty^2,$$

where the state-action pair (at stage $h$) $(s_h, a_h)$ is stochasticly generated from the induced distribution $d_h^\mu$.

**Remark 3.7.** Similar uniform coverage assumptions have also been considered in Wang et al. (2020a); Min et al. (2021); Yin et al. (2022a); Xiong et al. (2023); Yin et al. (2022b). Among these works, Yin et al. (2022b) is the most related to ours, proving an instance-dependent regret bound under general function approximation. Here we make a comparison with their assumption. In detail, Yin et al. (2022b) considered the differentiable function class, which is defined as follows

$$\mathcal{F} := \left\{ f\big(\boldsymbol{\theta}, \phi(\cdot, \cdot)\big) : \mathcal{S} \times \mathcal{A} \to \mathbb{R}, \boldsymbol{\theta} \in \Theta \right\}.$$

They introduced the following coverage assumption such that for all stage $h \in [H]$, there exists a constant $\kappa$,

$$\mathbb{E}_{d_h^\mu} \left[ \left( f(\boldsymbol{\theta}_1, \phi(s, a)) - f(\boldsymbol{\theta}_2, \phi(s, a)) \right)^2 \right] \geq \kappa \|\boldsymbol{\theta}_1 - \boldsymbol{\theta}_2\|_2^2, \forall \boldsymbol{\theta}_1, \boldsymbol{\theta}_2 \in \Theta; \quad (*)$$

$$\mathbb{E}_{d_h^\mu} \left[ \nabla f(\boldsymbol{\theta}, \phi(s, a)) \nabla f(\boldsymbol{\theta}, \phi(s, a))^\top \right] \succ \kappa I, \forall \boldsymbol{\theta} \in \Theta. \quad (**)$$

We can prove that our assumption is weaker than the first assumption (*), and we do not need the second assumption (**). This suggests that the differentiable function class studied in Yin et al. (2022b) is an example covered by our general function class.

In addition, in the special case of linear function class, the coverage assumption in Yin et al. (2022b) will reduce to the following linear function coverage assumption (Wang et al., 2020a; Min et al., 2021; Yin et al., 2022a; Xiong et al., 2023).

$$\lambda_{\min}(\mathbb{E}_{d_h^\mu}[\phi(s, a)\phi(s, a)^\top]) = \kappa > 0, \ \forall h \in [H].$$

Therefore, our assumption is also weaker than the linear function coverage assumption when dealing with the linear function class. Due to space limitations, we defer a detailed comparison to Appendix B.

**Remark 3.8.** Many works such as Uehara & Sun (2021); Xie et al. (2021a); Cheng et al. (2022); Ozdaglar et al. (2023); Rashidinejad et al. (2022); Zhu et al. (2023) adopted a weaker partial coverage assumption than ours, where the $\ell_\infty$ norm on the right hand is replaced with the expectation over a distribution corresponding to a single policy, typically the optimal one. Their assumption, however, generally confines their results to worst-case scenarios. It is unclear if we can still prove the instance-dependent regret under their assumption. We will explore it in the future.

## 4 ALGORITHM

In this section, we provide a comprehensive and detailed description of our algorithm (PNLSVI), as displayed in Algorithm 1. In the sequel, we introduce the key ideas of the proposed algorithm.

### 4.1 PESSIMISTIC VALUE ITERATION BASED PLANNING

Our algorithm operates in two distinct phases, the Variance Estimate Phase and the Pessimistic Planning Phase. At the beginning of the algorithm, the dataset is divided into two independent disjoint subsets $\mathcal{D}, \bar{\mathcal{D}}$ with equal size $K$, and each is assigned to a specific phase.

**Algorithm 1** Pessimistic Nonlinear Least-Squares Value Iteration (PNLSVI)

**Require:** Input confidence parameters $\bar{\beta}_h$, $\beta_h$ and $\epsilon > 0$.

1: **Initialize**: Split the input dataset into $\mathcal{D} = \{s_h^k, a_h^k, r_h^k\}_{k,h=1}^{K,H}$, $\bar{\mathcal{D}} = \{\bar{s}_h^k, \bar{a}_h^k, \bar{r}_h^k\}_{k,h=1}^{K,H}$ ; Set the value function $\widehat{f}_{H+1}(\cdot) = \check{f}_{H+1}(\cdot) = 0$.

2: //Constructing the variance estimator

3: **for** stage $h = H, \ldots, 1$ **do**

4:     $\bar{f}_h = \mathrm{argmin}_{f_h \in \mathcal{F}_h} \sum_{k \in [K]} \left( f_h(\bar{s}_h^k, \bar{a}_h^k) - \bar{r}_h^k - \check{f}_{h+1}(\bar{s}_{h+1}^k) \right)^2$.

5:     $\bar{g}_h = \mathrm{argmin}_{g_h \in \mathcal{F}_h} \sum_{k \in [K]} \left( g_h(\bar{s}_h^k, \bar{a}_h^k) - \left( \bar{r}_h^k + \check{f}_{h+1}(\bar{s}_{h+1}^k) \right)^2 \right)^2$.

6:     Calculate a bonus function $\bar{b}_h$ with confidence parameter $\bar{\beta}_h$,

7:     $\check{f}_h \leftarrow \{\bar{f}_h - \bar{b}_h - \epsilon\}_{[0, H-h+1]}$;

8:     Construct the variance estimator
$$\widehat{\sigma}_h^2(s, a) = \max \left\{ 1, \bar{g}_h(s,a) - (\bar{f}_h(s,a))^2 - O\left( \frac{\sqrt{\log(\mathcal{N} \cdot \mathcal{N}_b)} H^3}{\sqrt{K}\kappa} \right) \right\}.$$

9: **end for**

10: //Pessimistic value iteration based planning

11: **for** stage $h = H, \ldots, 1$ **do**

12:     $\widetilde{f}_h = \mathrm{argmin}_{f_h \in \mathcal{F}_h} \sum_{k \in [K]} \frac{1}{\widehat{\sigma}_h^2(s_h^k, a_h^k)} \left( f_h(s_h^k, a_h^k) - r_h^k - \widehat{f}_{h+1}(s_{h+1}^k) \right)^2$

13:     Calculate a bonus function $b_h$ with bonus parameter $\beta_h$;

14:     $\widehat{f}_h \leftarrow \{\widetilde{f}_h - b_h - \epsilon\}_{[0, H-h+1]}$;

15:     $\widehat{\pi}_h(\cdot|s) = \mathrm{argmax}_a \widehat{f}_h(s, a)$.

16: **end for**

17: **Output:** $\widehat{\pi} = \{\widehat{\pi}_h\}_{h=1}^H$.

The basic framework of our algorithm follows the pessimistic value iteration, which was initially introduced by Jin et al. (2021b). In details, for each stage $h \in [H]$, we construct the estimator value function $\widetilde{f}_h$ by solving the following variance-weighted ridge regression (Line 13):

$$\widetilde{f}_h = \mathrm{argmin}_{f_h \in \mathcal{F}_h} \sum_{k \in [K]} \frac{1}{\widehat{\sigma}_h^2(s_h^k, a_h^k)} \left( f_h(s_h^k, a_h^k) - r_h^k - \widehat{f}_{h+1}(s_{h+1}^k) \right)^2,$$

where $\widehat{\sigma}_h^2$ is the estimated variance and will be discussed in Section 4.2. In Line 14, we subtract the confidence bonus function $b_h$ from the estimator value function $\widetilde{f}_h$ to construct the pessimistic value function $\widehat{f}_h$. With the help of the confidence bonus function $b_h$, the pessimistic value function $\widehat{f}_h$ is almost a lower bound for the optimal value function $f_h^*$. The details of the bonus function will be discussed in Section 4.3.

Based on the pessimistic value function $\widehat{f}_h$ for horizon $h$, we recursively perform the value iteration for the horizon $h - 1$. Finally, we use the pessimistic value function $\widehat{f}_h$ to do planning and output the greedy policy with respect to the pessimistic value function $\widehat{f}_h$ (Lines 15 - 17).

4.2   VARIANCE ESTIMATOR

In this phase, we provide an estimator for the variance $\widehat{\sigma}_h$ in the weighted ridge regression. We construct this variance estimator with $\bar{\mathcal{D}}$, thus independent of $\mathcal{D}$. Using a larger bonus function $\bar{b}_h$, we conduct a pessimistic value iteration process similar to that discussed in Section 4.1 and obtain a more crude estimated value function $\{\check{f}_h\}_{h \in [H]}$. According to the definition of Bellman operators $\mathcal{T}$ and $\mathcal{T}_2$, the variance of the function $\check{f}_{h+1}$ for each state-action pair $(s, a)$ can be denoted by

$$[\mathrm{Var}_h \check{f}_{h+1}](s, a) = [\mathcal{T}_{2,h} \check{f}_{h+1}](s, a) - \left( [\mathcal{T}_h \check{f}_{h+1}](s, a) \right)^2.$$

Therefore, we need to estimate the first-order and second-order moments for $\check{f}_{h+1}$. We perform nonlinear least-squares regression separately for each of these moments. Specifically, in Line 4, we conduct regression to estimate the first-order moment.

$$\bar{f}_h = \mathrm{argmin}_{f_h \in \mathcal{F}_h} \sum_{k \in [K]} \left( f_h(\bar{s}_h^k, \bar{a}_h^k) - \bar{r}_h^k - \check{f}_{h+1}(\bar{s}_{h+1}^k) \right)^2.$$

In Line 5, we perform regression for estimating the second-order moment.

$$\bar{g}_h = \operatorname*{argmin}_{g_h \in \mathcal{F}_h} \sum_{k \in [K]} \left( g_h(\bar{s}_h^k, \bar{a}_h^k) - \left( \bar{r}_h^k + \breve{f}_{h+1}(\bar{s}_{h+1}^k) \right)^2 \right)^2 .$$

In this phase, we set the variance function to 1 for each state-action pair $(s,a)$. Combing these two regression results and subtracting some perturbing terms (We will discuss in Section 6.1), we create a pessimistic estimator for the variance function (Lines 7 to 8).

### 4.3 NONLINEAR BONUS FUNCTION

As we discuss in Sections 4.1 and 4.2, following Wang et al. (2020b); Kong et al. (2021); Agarwal et al. (2023), we introduce a bonus function. This function is designed to account for the functional uncertainty, enabling us to develop a pessimistic estimate of the value function. Ideally, we hope to choose $b_h(\cdot, \cdot) = \beta_h D_{\mathcal{F}_h}(\cdot, \cdot; \mathcal{D}_h; \widehat{\sigma}_h^2)$, where $\beta_h$ is the confidence parameter and $D_{\mathcal{F}_h}(\cdot, \cdot; \mathcal{D}_h; \widehat{\sigma}_h^2)$ is defined in Definition 3.4. However, the $D^2$-divergence composes a complex function class, and its calculation involves solving a complex optimization problem. To address this issue, following Agarwal et al. (2023), we assume there exists a function class $\mathcal{W}$ with cardinally $|\mathcal{W}| = \mathcal{N}_b$ and can approximate the $D^2$-divergence well. For the parameters $\beta_h, \lambda \geq 0$, error parameter $\epsilon \geq 0$, taking a variance function $\sigma_h(\cdot, \cdot) : \mathcal{S} \times \mathcal{A} \to \mathbb{R}$ and $\widehat{f}_h \in \mathcal{F}_h$ as input, we can get a bonus function $b_h(\cdot, \cdot) \in \mathcal{W}$ satisfying the following properties:

- $b_h(z_h) \geq \max \left\{ |f_h(z_h) - \widehat{f}_h(z_h)|, f_h \in \mathcal{F}_h : \sum_{k \in [K]} \frac{(f_h(z_h^k) - \widehat{f}_h(z_h^k))^2}{(\widehat{\sigma}_h(s_h^k, a_h^k))^2} \leq (\beta_h)^2 \right\}$ for any $z_h \in \mathcal{S} \times \mathcal{A}$.

- $b_h(z_h) \leq C \cdot \left( D_{\mathcal{F}_h}(z_h; \mathcal{D}_h; \widehat{\sigma}_h^2) \cdot \sqrt{(\beta_h)^2 + \lambda} + \epsilon \beta_h \right)$ for all $z_h \in \mathcal{S} \times \mathcal{A}$ with constant $0 < C < \infty$.

We can implement the function class $\mathcal{W}$ with bounded $\mathcal{N}_b$ by extending the online subsampling framework presented in Wang et al. (2020b); Kong et al. (2021) to an offline dataset. Additionally, using Algorithm 1 of Kong et al. (2021), we can calculate the bonus function with finite calls of the oracle for solving the variance-weighted ridge regression problem. We leave a detailed discussion to Appendix C.

## 5 MAIN RESULTS

In this section, we prove an instance-dependent regret bound of Algorithm 1.

**Theorem 5.1.** Under Assumption 3.6, for $K \geq \widetilde{\Omega}\left( \frac{\log(\mathcal{N} \cdot \mathcal{N}_b) H^6}{\kappa^2} \right)$, if we set the parameters $\beta'_{1,h}, \beta'_{2,h} = \widetilde{O}(\sqrt{\log(\mathcal{N} \cdot \mathcal{N}_b)} H^2)$ and $\beta_h = \widetilde{O}(\sqrt{\log \mathcal{N}})$ in Algorithm 1, then with probability at least $1 - \delta$, for any state $s \in \mathcal{S}$, we have

$$V_1^*(s) - V_1^{\widehat{\pi}}(s) \leq \widetilde{O}(\sqrt{\log \mathcal{N}}) \sum_{h=1}^{H} \mathbb{E}_{\pi^*} \left[ D_{\mathcal{F}_h}(z_h; \mathcal{D}_h; [\mathbb{V}_h V_{h+1}^*](\cdot, \cdot)) | s_1 = s \right],$$

where $[\mathbb{V}_h V_{h+1}^*](s, a) = \max\{1, [\operatorname{Var}_h V_{h+1}^*](s, a)\}$ is the truncated conditional variance.

This theorem establishes an upper bound for the suboptimality of our policy $\widehat{\pi}$. The bound depends on the expected uncertainty, which is characterized by the weighted $D^2$-divergence along the trajectory, marking itself as an instance-dependent result. It's noteworthy that both the trajectory and the weight function are based on the optimal policy and the optimal value function, respectively. This bound necessitates that the dataset size $K$ is sufficiently large. Furthermore, all parameters are determined solely by the complexity of the function class, the horizon length $H$, and the data coverage assumption constant $\kappa$, regardless of the dataset's composition.

**Remark 5.2.** In Theorem 5.1, the dependence on the cardinality of the function class scales as $\widetilde{O}(\sqrt{\log \mathcal{N}})$, which is better than that in Yin et al. (2022b), which scales as $\widetilde{O}(\log \mathcal{N})$. This improved dependence is due to the reference-advantage decomposition (discussed in Section 6.2), which avoids the unnecessary covering argument. Thus we resolve the open problem raised by Yin et al. (2022b), i.e., how to achieve the $\widetilde{O}(\sqrt{\log \mathcal{N}})$ dependence.

**Remark 5.3.** When specialized to linear MDPs (Jin et al., 2020), the following function class

$$\mathcal{F}_h^{\text{lin}} = \{ \langle \phi(\cdot, \cdot), \boldsymbol{\theta}_h \rangle : \boldsymbol{\theta}_h \in \mathbb{R}^d, \|\boldsymbol{\theta}_h\|_2 \leq B_h \} \text{ for any } h \in [H],$$

suffices and satisfies the completeness assumption (Assumption 3.3). Let $\mathcal{F}_h^{\text{lin}}(\epsilon)$ be an $\epsilon$-net of the linear function class $\mathcal{F}_h^{\text{lin}}$, with $\log |\mathcal{F}_h^{\text{lin}}(\epsilon)| = \widetilde{O}(d)$. The dependency of the function class will reduce to $\widetilde{O}(\sqrt{\log \mathcal{N}}) = \widetilde{O}(\sqrt{d})$. For linear function class, we can prove the following inequality:

$$D_{\mathcal{F}_h^{\text{lin}}(\epsilon)}(z; \mathcal{D}_h; [\mathbb{V}_h V_{h+1}^*](\cdot, \cdot)) \leq \|\phi(z)\|_{\Sigma_h^{*-1}},$$

where $\Sigma_h^* = \sum_{k \in [K]} \phi(s_h^k, a_h^k) \phi(s_h^k, a_h^k)^\top / [\mathbb{V}_h V_{h+1}^*](s_h^k, a_h^k) + \lambda \mathbf{I}$. Therefore, our regret guarantee in Theorem 5.1 is reduced to

$$V_1^*(s) - V_1^{\widehat{\pi}}(s) \le \widetilde{O}(\sqrt{d}) \cdot \sum_{h=1}^{H} \mathbb{E}_{\pi^*} \left[ \|\phi(s_h, a_h)\|_{\mathbf{\Sigma}_h^{*-1}} | s_1 = s \right],$$

which matches the lower bound proved in Xiong et al. (2023). This suggests that our algorithm is optimal for linear MDPs.

## 6 KEY TECHNIQUES

In this section, we provide an overview of the key techniques in our algorithm design and analysis.

### 6.1 VARIANCE ESTIMATOR WITH NONLINEAR FUNCTION CLASS

In our work, we extend the technique of variance-weighted ridge regression, first introduced in Zhou et al. (2021) for online RL, and later used by Min et al. (2021); Yin et al. (2022a;b); Xiong et al. (2023) for offline RL with linear MDPs, to general nonlinear function class $\mathcal{F}$. We use the following nonlinear least-squares regression to estimate the underlying value function:

$$\widetilde{f}_h = \underset{f_h \in \mathcal{F}_h}{\operatorname{argmin}} \sum_{k \in [K]} \frac{1}{\widehat{\sigma}_h^2(s_h^k, a_h^k)} \left( f_h(s_h^k, a_h^k) - r_h^k - \widehat{f}_{h+1}(s_{h+1}^k) \right)^2.$$

For this regression, it is crucial to obtain a reliable evaluation for the variance of the estimated cumulative reward $r_h^k + \widehat{f}_{h+1}(s_{h+1}^k)$. As discussed in Section 4.2, we use $\bar{\mathcal{D}}$ to construct a variance estimator independent from $\mathcal{D}$. According to the definition of Bellman operators $\mathcal{T}$ and $\mathcal{T}_2$, the variance of the function $\breve{f}_{h+1}$ for each state-action pair $(s, a)$ can be denoted by

$$[\operatorname{Var}_h \breve{f}_{h+1}](s, a) = [\mathcal{T}_{2,h} \breve{f}_{h+1}](s, a) - \left( [\mathcal{T}_h \breve{f}_{h+1}](s, a) \right)^2.$$

In our algorithm, we perform nonlinear least-squares regression on $\bar{\mathcal{D}}$. For simplicity, we denote the empirical variance as $\mathbb{B}_h(s, a) = \bar{g}_h(s, a) - \left( \bar{f}_h(s, a) \right)^2$, and the difference between empirical variance $\mathbb{B}_h(s, a)$ and actual variance $[\operatorname{Var}_h \breve{f}_{h+1}](s, a)$ is upper bound by

$$\left| \mathbb{B}_h(s, a) - [\operatorname{Var}_h \breve{f}_{h+1}](s, a) \right| \le \left| \bar{g}_h(s, a) - [\mathcal{T}_{2,h} \breve{f}_{h+1}](s, a) \right| + \left| \left( \bar{f}_h(s, a) \right)^2 - \left( [\mathcal{T}_h \breve{f}_{h+1}](s, a) \right)^2 \right|.$$

For these nonlinear function estimators, the following lemmas provide coarse concentration properties for the first and second order Bellman operators.

**Lemma 6.1.** For any stage $h \in [H]$, let $\breve{f}_{h+1}(\cdot, \cdot) \le H$ be the estimated value function constructed in Algorithm 1 Line 7. By utilizing Assumption 3.3, there exists a function $\bar{f}'_h \in \mathcal{F}_h$, satisfying that $|\bar{f}'_h(z_h) - [\mathcal{T}_h \breve{f}_{h+1}](z_h)| \le \epsilon$ holds for all state-action pair $z_h = (s_h, a_h)$. Then with probability at least $1 - \delta/(4H)$, it holds that

$$\sum_{k \in [K]} \left( \bar{f}'_h(\bar{z}_h^k) - \bar{f}_h(\bar{z}_h^k) \right)^2 \le (\bar{\beta}_{1,h})^2,$$

where $\bar{\beta}_{1,h} = \widetilde{O} \left( \sqrt{\log(\mathcal{N} \cdot \mathcal{N}_b)} H \right)$, and $\bar{f}_h$ is the estimated function for first-moment Bellman operator (Line 4 in Algorithm 1).

**Lemma 6.2.** For any stage $h \in [H]$, let $\breve{f}_{h+1}(\cdot, \cdot) \le H$ be the estimated value function constructed in Algorithm 1 Line 7. By utilizing Assumption 3.3, there exists a function $\bar{g}'_h \in \mathcal{F}_h$, satisfying that $|\bar{g}'_h(z_h) - [\mathcal{T}_{2,h} \breve{f}_{h+1}](z_h)| \le \epsilon$ holds for all state-action pair $z_h = (s_h, a_h)$. Then with probability at least $1 - \delta/(4H)$, it holds that

$$\sum_{k \in [K]} \left( \bar{g}'_h(\bar{z}_h^k) - \bar{g}_h(\bar{z}_h^k) \right)^2 \le (\bar{\beta}_{2,h})^2,$$

where $\bar{\beta}_{2,h} = \widetilde{O} \left( \sqrt{\log(\mathcal{N} \cdot \mathcal{N}_b)} H^2 \right)$, and $\bar{g}_h$ is the estimated function for second-moment Bellman operator (Line 5 in Algorithm 1).

Notice that all of the previous analysis focuses on the estimated function $\breve{f}_{h+1}$. By leveraging an induction procedure similar to existing works in the linear case (Jin et al., 2021b), we can control the distance between the estimated function $\breve{f}_{h+1}$ and the optimal value function $f_{h+1}^*$. In details, with high

probability, for all stage $h \in [H]$, the distance is upper bounded by $\widetilde{O}\left(\sqrt{\log(\mathcal{N} \cdot \mathcal{N}_b)}H^3/\sqrt{K\kappa}\right)$.

This result allows us to further bound the difference between $[\mathrm{Var}_h \breve{f}_{h+1}](s,a)$ and $[\mathrm{Var}_h f_{h+1}^*](s,a)$. Therefore, the concentration properties in Lemmas 6.1 and 6.2 enable us to add some perturbation terms and construct a variance estimator, which satisfies the following property:

$$[\mathbb{V}_h V_{h+1}^*](s,a) - \widetilde{O}\left(\frac{\sqrt{\log(\mathcal{N} \cdot \mathcal{N}_b)}H^3}{\sqrt{K\kappa}}\right) \le \widehat{\sigma}_h^2(s,a) \le [\mathbb{V}_h V_{h+1}^*](s,a). \tag{6.1}$$

where $[\mathbb{V}_h V_{h+1}^*](s,a) = \max\{1, [\mathrm{Var}_h V_{h+1}^*](s,a)\}$ is the truncated conditional variance.

## 6.2 Reference-Advantage Decomposition

To obtain the optimal dependency on the function class complexity, we need to tackle the challenge of additional error from uniform concentration over the whole function class $\mathcal{F}_h$. To address this problem, we utilize the so-called reference-advantage decomposition technique, which was used by (Xiong et al., 2023) to achieve the optimal regret for offline RL with linear function approximation. We generalize this technique to nonlinear function approximation. We provide detailed insights into this approach as follows:

$$r_h(s_h, a_h) + \widehat{f}_{h+1}(s_{h+1}) - [\mathcal{T}_h \widehat{f}_{h+1}](s_h, a_h) = \underbrace{r_h(s_h, a_h) + f_{h+1}^*(s_{h+1}) - [\mathcal{T}_h f_{h+1}^*](s_h, a_h)}_{\text{Reference uncertainty}}$$

$$+ \underbrace{\widehat{f}_{h+1}(s_{h+1}) - f_{h+1}^*(s_{h+1}) - ([\mathbb{P}_h \widehat{f}_{h+1}](s_h, a_h) - [\mathbb{P}_h f_{h+1}^*](s_h, a_h))}_{\text{Advantage uncertainty}}.$$

We decompose the Bellman error into two parts: the Reference uncertainty and the Advantage uncertainty. For the first term, the optimal value function $f_{h+1}^*$ is fixed and not related to the pre-collected dataset, which circumvents additional uniform concentration over the whole function class and avoids any dependence on the function class complexity. For the second term, it is worth noticing that the distance between the estimated function $\widehat{f}_{h+1}$ and the optimal value function $f_h^*$ decreases with the speed of $O(1/\sqrt{K\kappa})$. Though, we still need to maintain the uniform convergence guarantee, the Advantage uncertainty is dominated by the Reference uncertainty when the number of episodes $K$ is large enough. Such an analysis approach has been first studied in the online RL setting (Azar et al., 2017; Zhang et al., 2021; Hu et al., 2022; He et al., 2022; Agarwal et al., 2023) and later in the offline environment by Xiong et al. (2023). Previous works, such as Yin et al. (2022b), didn't adapt the reference-advantage decomposition analysis to their nonlinear function class, resulting in a parameter space dependence that scales with $d$, instead of the optimal $\sqrt{d}$. By integrating these results, we can prove a variance-weighted concentration inequality for Bellman operators.

**Lemma 6.3.** For each stage $h \in [H]$, assuming the variance estimator $\widehat{\sigma}_h$ satisfies (6.1), let $\widehat{f}_{h+1}(\cdot, \cdot) \le H$ be the estimated value function constructed in Algorithm 1 Line 14. Then with probability at least $1 - \delta/(4H)$, it holds that $\sum_{k \in [K]} \frac{1}{(\widehat{\sigma}_h(z_h^k))^2}\left([\mathcal{T}_h \widehat{f}_{h+1}](z_h^k) - \widetilde{f}_h(z_h^k)\right)^2 \le (\beta_h)^2$, where $\beta_h = \widetilde{O}(\sqrt{\log \mathcal{N}})$ and $\widetilde{f}_h$ is the estimated function from the weighted ridge regression (Line 12 in Algorithm 1).

After controlling the Bellman error, with a similar argument to Jin et al. (2021b), we obtain the following lemma, which provides an upper bound for the regret.

**Lemma 6.4** (Regret Decomposition Property). If $\left|[\mathcal{T}_h \widehat{f}_{h+1}](z) - \widetilde{f}_h(z)\right| \le b_h(z)$ holds for all stage $h \in [H]$ and state-action pair $z = (s,a) \in \mathcal{S} \times \mathcal{A}$, then the suboptimality of the output policy $\widehat{\pi}$ in Algorithm 1 can be bounded as

$$V_1^*(s) - V_1^{\widehat{\pi}}(s) \le 2\sum_{h=1}^H \mathbb{E}_{\pi^*}\left[b_h(s_h, a_h) \mid s_1 = s\right].$$

Here, the expectation $\mathbb{E}_{\pi^*}$ is with respect to the trajectory induced by $\pi^*$ in the underlying MDP.

Combining the results in Lemmas 6.3 and 6.4, we have proved Theorem 5.1.

## 7 Conclusion and Future Work

In this paper, we present Pessimistic Nonlinear Least-Square Value Iteration (PNLSVI), an oracle-efficient algorithm for offline RL with non-linear function approximation. Our result matches the lower bound proved in Xiong et al. (2023) when specialized to linear function approximation. We notice that our uniform coverage assumption can sometimes be strong in practice. In our future work, we plan to relax this assumption by devising algorithms for nonlinear function classes under a partial coverage assumption.

ACKNOWLEDGEMENTS

We thank the anonymous reviewers and area chair for their helpful comments. QD, HZ, JH and QG are supported in part by the National Science Foundation CAREER Award 1906169, CPS-2312094, Amazon Research Award and the Sloan Research Fellowship. JH is also supported in part by Amazon PhD Fellowship. The views and conclusions contained in this paper are those of the authors and should not be interpreted as representing any funding agencies.

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

# A COMPARISON OF OFFLINE RL ALGORITHMS

In this section, we make a comparison between different offline RL algorithms, concerning their algorithm type, function approximation class, data coverage assumption, used oracles, and regret type. In the upper part are the algorithms with worst-case regret, while in the lower part are those with instance-dependent regret.

Table 1: Comparison of offline RL algorithms in terms of algorithm type, function classes, data coverage assumption, types of oracle and regret-type.

| Algorithm | Algorithm Type | Function Classes | Data Coverage | Types of Oracle | Regret Type |
|---|---|---|---|---|---|
| Xie et al. (2021a) | Bellman-consistent Pessimism | General | Partial | Optimization on Policy and Function Class | Worst-case |
| CPPO-TV Uehara & Sun (2021) | MLE | General | Partial | Optimization on Policy and Hypothesis Class | Worst-case |
| CORAL Rashidinejad et al. (2022) | Augmented Lagrangian with MIS | General | Partial | Optimization on Policy and Function Class | Worst-case |
| Reformulated LP Ozdaglar et al. (2023) | Linear Program | General | Partial | Linear Programming | Worst-case |
| ATAC Cheng et al. (2022) | Actor Critic | General | Partial | No-regret Policy Optimization and Optimization on the Function Class | Worst-case |
| A-Crab Zhu et al. (2023) | Actor Critic | General | Partial | No-regret Policy Optimization and Optimization on the Function Class | Worst-case |
| LinPEVI-ADV+ Xiong et al. (2023) | LSVI-type | Linear | Uniform | / | Instance-dependent |
| PFQL Yin et al. (2022b) | LSVI-type | Differentible | Uniform | Gradient Oracle and Optimization on the Function Class | Instance-dependent |
| PNLSVI (Our work) | LSVI-type | General | Uniform | Regression Oracle | Instance-dependent |

# B COMPARISON OF DATA COVERAGE ASSUMPTIONS

In Yin et al. (2022b), they studied the general differentiable function class, where the function class can be denoted by

$$\mathcal{F} := \Big\{ f\big(\boldsymbol{\theta}, \boldsymbol{\phi}(\cdot, \cdot)\big) : \mathcal{X} \times \mathcal{A} \to \mathbb{R}, \boldsymbol{\theta} \in \Theta \Big\}.$$

In this definition, $\Psi$ is a compact subset, and $\boldsymbol{\phi}(\cdot, \cdot) : \mathcal{X} \times \mathcal{A} \to \Psi \subseteq \mathbb{R}^m$ is a feature map. The parameter space $\Theta$ is a compact subset $\Theta \subseteq \mathbb{R}^d$. The function $f : \mathbb{R}^d \times \mathbb{R}^m \to \mathbb{R}$ satisfies the following smoothness conditions:

- For any vector $\boldsymbol{\phi} \in \mathbb{R}^m$, $f(\boldsymbol{\theta}, \boldsymbol{\phi})$ is third-time differentiable with respect to the parameter $\boldsymbol{\theta}$.
- Functions $f, \partial_{\boldsymbol{\theta}} f, \partial^2_{\boldsymbol{\theta}, \boldsymbol{\theta}} f, \partial^3_{\boldsymbol{\theta}, \boldsymbol{\theta}, \boldsymbol{\theta}} f$ are jointly continuous for $(\boldsymbol{\theta}, \boldsymbol{\phi})$.

Under this definition, Yin et al. (2022b) introduce the following coverage assumption (Assumption 2.3) such that for all stage $h \in [H]$, there exists a constant $\kappa$,

$$\mathbb{E}_{d_h^\mu} \Big[ \big( f(\boldsymbol{\theta}_1, \boldsymbol{\phi}(x, a)) - f(\boldsymbol{\theta}_2, \boldsymbol{\phi}(x, a)) \big)^2 \Big] \geq \kappa \|\boldsymbol{\theta}_1 - \boldsymbol{\theta}_2\|_2^2, \forall \boldsymbol{\theta}_1, \boldsymbol{\theta}_2 \in \Theta; \; (*)$$
$$\mathbb{E}_{d_h^\mu} \big[ \nabla f(\boldsymbol{\theta}, \boldsymbol{\phi}(x, a)) \nabla f(\boldsymbol{\theta}, \boldsymbol{\phi}(x, a))^\top \big] \succ \kappa I, \forall \boldsymbol{\theta} \in \Theta. \; (**)$$

It is worth noting that our assumption 3.6 is weaker than this assumption. For any compact sets $\Theta, \Psi$ and continuous function $f$, there always exist a constant $\kappa_0 > 0$ such that $f$ is $\kappa_0$-Lipschitz with respect to the parameter $\boldsymbol{\theta}$,i.e:

$$|f(\boldsymbol{\theta}_1, \boldsymbol{\phi}) - f(\boldsymbol{\theta}_2, \boldsymbol{\phi})| \leq \kappa_0 \|\boldsymbol{\theta}_1 - \boldsymbol{\theta}_2\|_2, \forall \boldsymbol{\theta}_1, \boldsymbol{\theta}_2 \in \Theta, \boldsymbol{\phi} \in \Psi.$$

Therefore, the coverage assumption in Yin et al. (2022b) implies that

$$\mathbb{E}_{d_h^\mu}\left[\left(f(\boldsymbol{\theta}_1, \boldsymbol{\phi}(\cdot, \cdot)) - f(\boldsymbol{\theta}_2, \boldsymbol{\phi}(\cdot, \cdot))\right)^2\right] \geq \kappa \|\boldsymbol{\theta}_1 - \boldsymbol{\theta}_2\|_2^2$$
$$\geq \frac{\kappa}{\kappa_0^2} \sup_{(x,a) \in \mathcal{X} \times \mathcal{A}} \left(f(\boldsymbol{\theta}_1, \boldsymbol{\phi}(x, a)) - f(\boldsymbol{\theta}_2, \boldsymbol{\phi}(x, a))\right)^2.$$

Our assumption can be reduced to their first assumption (*). We do not need the second assumption (**).

In addition, for the linear function class, the coverage assumption in Yin et al. (2022b) will reduce to the following linear function coverage assumption(Wang et al., 2020a; Min et al., 2021; Yin et al., 2022a; Xiong et al., 2023).

$$\lambda_{\min}(\mathbb{E}_{\mu,h}[\phi(x,a)\phi(x,a)^\top]) = \kappa > 0, \ \forall h \in [H].$$

Therefore, our assumption is also weaker than the linear function coverage assumption when dealing with the linear function class.

## C  DISCUSSION ON THE CALCULATION OF THE BONUS FUNCTION

To obtain the bonus function satisfying the required properties, we need to consider the constraint optimization problem

$$\max\left\{|f_1(z_h) - f_2(z_h)|, f_1, f_2 \in \mathcal{F}_h : \sum_{k \in [K]} \frac{(f_1(z_h^k) - f_2(z_h^k))^2}{(\widehat{\sigma}_h(s_h^k, a_h^k))^2} \leq (\beta_h)^2\right\}. \tag{C.1}$$

We utilize the online subsampling framework presented in Wang et al. (2020b); Kong et al. (2021) to an offline dataset. Additionally, we generalize the idea of Kong et al. (2021) to the weighted problem. Using the shorthand expression

$$\|f\|_{\widehat{\sigma}_h, \mathcal{D}_h}^2 = \sum_{k \in [K]} \frac{(f(z_h^k))^2}{(\widehat{\sigma}_h(s_h^k, a_h^k))^2},$$

we aim to estimate the solution of the following constraint optimization problem.

$$\min_{f_1, f_2} \|f_1 - f_2\|_{\widehat{\sigma}_h, \mathcal{D}_h}^2 + \frac{w}{2}(f_1(s,a) - f_2(s,a) - 2L)^2. \tag{C.2}$$

---

**Algorithm 2** Binary Search

1: **Input:** Dataset $\mathcal{D}_h = \{s_h^k, a_h^k, r_h^k\}_{k=1}^K$, objective $z = (s,a)$, $\beta_h$, precision $\alpha$
2: $\mathcal{G} \leftarrow \mathcal{F}_h - \mathcal{F}_h$
3: Define $R(g, w) := \|g\|_{\widehat{\sigma}_h, \mathcal{D}_h}^2 + \frac{w}{2}(g(s,a) - 2(L+1))^2, \forall g \in \mathcal{G}$
4: $w_L \leftarrow 0, w_H \leftarrow \beta_h/(\alpha(L+1))$
5: $g_L \leftarrow 0, z_L \leftarrow 0$
6: $g_H \leftarrow \arg\min_{g \in \mathcal{G}} R(g, w_H), z_H \leftarrow g_H(s,a)$
7: $\Delta \leftarrow \alpha\beta/(8(L+1)^3)$
8: **while** $|z_H - z_L| > \alpha$ and $|w_H - w_L| > \Delta$ **do**
9:   $\widetilde{w} \leftarrow (w_H + w_L)/2$
10:   $\widetilde{g} \leftarrow \arg\min_{g \in \mathcal{G}} R(g, \widetilde{w}), \widetilde{z} \leftarrow \widetilde{g}(s,a)$
11:   **if** $\|\widetilde{g}\|_{\widehat{\sigma}_h, \mathcal{D}_h}^2 > \beta_h$ **then**
12:     $w_H \leftarrow \widetilde{w}, z_H \leftarrow \widetilde{z}$
13:   **else**
14:     $w_L \leftarrow \widetilde{w}, z_L \leftarrow \widetilde{z}$
15:   **end if**
16: **end while**
17: **Output:** $z_H$

---

The binary search algorithm is similar to Kong et al. (2021), which utilizes the oracle for variance-weighted ridge regression in Lines 6 and 10. It reduces the computational complexity of the constrained optimization problem given by (C.1), limiting it to a finite number of calls to the regression oracle. If the function class $\mathcal{F}_h$ is convex, the binary search algorithm can solve the optimization problem (C.1) up to a precision of $\alpha$ with $O(\log(1/\alpha))$ calls of the regression oracle. We summarize the result in the following theorem. If $\mathcal{F}_h$ is not convex, the method of Krishnamurthy et al. (2017) can solve the problem with $O(1/\alpha)$ calls of the regression oracle.

**Theorem C.1.** Assume the optimal solution of the optimization problem (C.1) is $g^* = f_1^* - f_2^*$ and the function class $\mathcal{F}_h$ is convex and closed under pointwise convergence, then Algorithm 2 terminate after $O(\log(1/\alpha))$ calls of the regression oracle and the returned values satisfy

$$|z_H - g^*(s,a)| \leq \alpha.$$

*Proof of Theorem C.1.* Easy to see $\mathcal{G} = \mathcal{F}_h - \mathcal{F}_h$ is also convex. We then follow the proof of Theorem 1 of Foster et al. (2018).

□

To see the solution of optimization problem C.1 satisfies the condition of the bonus function, we recall the definition of the $D^2$-divergence

$$D^2_{\mathcal{F}_h}(z; \mathcal{D}_h; \widehat{\sigma}_h^2) = \sup_{f_1, f_2 \in \mathcal{F}_h} \frac{(f_1(z) - f_2(z))^2}{\sum_{k \in [K]} \frac{1}{(\widehat{\sigma}_h(z_h^k))^2}(f_1(z_h^k) - f_2(z_h^k))^2 + \lambda}.$$

Therefore, for any $f_1, f_2 \in \mathcal{F}_h$ satisfying $\sum_{k \in [K]} \frac{1}{(\widehat{\sigma}_h(z_h^k))^2}(f_1(z_h^k) - f_2(z_h^k))^2 \leq (\beta_h^2)$, we have

$$|f_1(z_h) - f_2(z_h)| \leq D_{\mathcal{F}_h}(z; \mathcal{D}_h; \widehat{\sigma}_h^2)\sqrt{(\beta_h)^2 + \lambda}.$$

## D   ANALYSIS OF THE VARIANCE ESTIMATOR

In this section, our main objective is to prove that our variance estimators are close to the truncated variance of the optimal value function $[\mathbb{V}_h V_{h+1}^*](s,a)$. The following lemmas are helpful in the proof of Lemma D.4. To start with, we need an upper bound of the $D^2$-divergence for a large dataset.

**Lemma D.1.** Let $\mathcal{D}_h$ be the dataset satisfying Assumption 3.6. When the size of data set satisfies $K \geq \widetilde{\Omega}\left(\frac{\log \mathcal{N}}{\kappa^2}\right)$, with probability at least $1 - \delta$, for each state-action pair $z$, we have

$$D_{\mathcal{F}_h}(z, \mathcal{D}_h, 1) = \widetilde{O}\left(\frac{1}{\sqrt{K\kappa}}\right).$$

Lemma 6.1 and Lemma 6.2 show the confidence radius for the first and second-order Bellman error, which is essential in our proof. Here we restate them with more accurate parameter choices.

**Lemma D.2** (Restatement of Lemma 6.1)**.** For any stage $h \in [H]$, let $\check{f}_{h+1}(\cdot, \cdot) \leq H$ be the estimated value function constructed in Algorithm 1 Line 7. By utilizing Assumption 3.3, there exists a function $\bar{f}_h' \in \mathcal{F}_h$, satisfying that $|\bar{f}_h'(z_h) - [\mathcal{T}_h \check{f}_{h+1}](z_h)| \leq \epsilon$ holds for all state-action pair $z_h = (s_h, a_h)$. Then with probability at least $1 - \delta/(4H)$, it holds that

$$\sum_{k \in [K]} \left(\bar{f}_h'(\bar{z}_h^k) - \bar{f}_h(\bar{z}_h^k)\right)^2 \leq (\bar{\beta}_{1,h})^2,$$

where $\bar{\beta}_{1,h} = \widetilde{O}\left(\sqrt{\log(\mathcal{N} \cdot \mathcal{N}_b)}H\right)$, and $\bar{f}_h$ is the estimated function for first-moment Bellman operator (Line 4 in Algorithm 1).

**Lemma D.3** (Restatement of Lemma 6.1)**.** For any stage $h \in [H]$, let $\check{f}_{h+1}(\cdot, \cdot) \leq H$ be the estimated value function constructed in Algorithm 1 Line 7. By utilizing Assumption 3.3, there exists a function $\bar{g}_h' \in \mathcal{F}_h$, satisfying that $|\bar{g}_h'(z_h) - [\mathcal{T}_{2,h}\check{f}_{h+1}](z_h)| \leq \epsilon$ holds for all state-action pair $z_h = (s_h, a_h)$. Then with probability at least $1 - \delta/(4H)$, it holds that

$$\sum_{k \in [K]} \left(\bar{g}_h'(\bar{z}_h^k) - \bar{g}_h(\bar{z}_h^k)\right)^2 \leq (\bar{\beta}_{2,h})^2,$$

where $\bar{\beta}_{2,h} = \widetilde{O}\left(\sqrt{\log(\mathcal{N} \cdot \mathcal{N}_b)}H^2\right)$, and $\bar{g}_h$ is the estimated function for second-moment Bellman operator (Line 5 in Algorithm 1).

Recall the definition of the variance estimator.

$$\bar{f}_h = \underset{f_h \in \mathcal{F}_h}{\operatorname{argmin}} \sum_{k \in [K]} \left(f_h(\bar{s}_h^k, \bar{a}_h^k) - \bar{r}_h^k - \check{f}_{h+1}(\bar{s}_{h+1}^k)\right)^2$$

$$\bar{g}_h = \underset{g_h \in \mathcal{F}_h}{\operatorname{argmin}} \sum_{k \in [K]} \left(g_h(\bar{s}_h^k, \bar{a}_h^k) - \left(\bar{r}_h^k + \check{f}_{h+1}(\bar{s}_{h+1}^k)\right)^2\right)^2.$$

The variance estimator is defined as:

$$\widehat{\sigma}_h^2(s,a) := \max\left\{1, \bar{g}_h(s,a) - \left(\bar{f}_h(s,a)\right)^2 - \widetilde{O}\left(\frac{\sqrt{\log(\mathcal{N}\cdot\mathcal{N}_b)}H^3}{\sqrt{K\kappa}}\right)\right\}.$$

We can prove the following lemma:

**Lemma D.4.** with probability at least $1-\delta/2$, for any $h\in[H]$, the variance estimator designed above satisfies:

$$[\mathbb{V}_h V_{h+1}^*](s,a) - \widetilde{O}\left(\frac{\sqrt{\log(\mathcal{N}\cdot\mathcal{N}_b)}H^3}{\sqrt{K\kappa}}\right) \le \widehat{\sigma}_h^2(s,a) \le [\mathbb{V}_h V_{h+1}^*](s,a).$$

*Proof of Lemma D.4.* We write $\mathbb{B}_h(s,a) = \bar{g}_h(s,a) - \left(\bar{f}_h(s,a)\right)^2$. We first bound the difference between $\mathbb{B}_h(s,a)$ and $[\mathrm{Var}_h \check{f}_{h+1}](s,a)$. By the definition of conditional variance, we have

$$\left|\mathbb{B}_h(s,a) - [\mathrm{Var}_h \check{f}_{h+1}](s,a)\right| \le \left|\bar{g}_h(s,a) - [\mathcal{T}_{2,h}\check{f}_{h+1}](s,a)\right| + \left|\left(\bar{f}_h(s,a)\right)^2 - \left([\mathcal{T}_h\check{f}_{h+1}](s,a)\right)^2\right|,$$

where we use our definition of Bellman operators. By Assumption 3.3, there exists $\bar{f}_h' \in \mathcal{F}_h, \bar{g}_h' \in \mathcal{F}_h$, such that for all $(s,a)$

$$\left|\bar{f}_h'(s,a) - [\mathcal{T}_h\check{f}_{h+1}](s,a)\right| \le \epsilon \tag{D.1}$$

$$\left|\bar{g}_h'(s,a) - [\mathcal{T}_{2,h}\check{f}_{h+1}](s,a)\right| \le \epsilon. \tag{D.2}$$

Then by Lemma D.2, with probability at least $1-\delta/(4H^2)$, the following inequality holds

$$\sum_{k\in[K]} \left(\bar{f}_h(\bar{z}_h^k) - \bar{f}_h'(\bar{z}_h^k)\right)^2 \le (\bar{\beta}_{1,h})^2. \tag{D.3}$$

Similarly, for the second-order term, using Lemma D.3, with probability at least $1-\delta/(4H^2)$, the following inequality holds

$$\sum_{k\in[K]} \left(\bar{g}_h(\bar{z}_h^k) - \bar{g}_h'(\bar{z}_h^k)\right)^2 \le (\bar{\beta}_{2,h})^2. \tag{D.4}$$

After taking a union bound, with probability at least $1-\delta/(2H)$, (D.3) and (D.4) hold for all $h\in[H]$ simultaneously. Consequently, we focus on this high probability event and prove that

$$\left|\bar{g}_h(s,a) - [\mathcal{T}_{2,h}\check{f}_{h+1}](s,a)\right| + \left|\left(\bar{f}_h(s,a)\right)^2 - \left([\mathcal{T}_h\check{f}_{h+1}](s,a)\right)^2\right|$$

$$\le \epsilon + \left|\bar{g}_h(s,a) - \bar{g}_h'(s,a)\right| + O(H)\cdot\left[\left|\bar{f}_h(s,a) - \bar{f}_h'(s,a)\right| + \epsilon\right]$$

$$= O(H)\cdot\epsilon + \frac{\left|\bar{g}_h(s,a) - \bar{g}_h'(s,a)\right|}{\sqrt{\sum_{k\in[K]}\left(\bar{g}_h(\bar{z}_h^k) - \bar{g}_h'(\bar{z}_h^k)\right)^2 + \lambda}}\cdot\sqrt{\sum_{k\in[K]}\left(\bar{g}_h(\bar{z}_h^k) - \bar{g}_h'(\bar{z}_h^k)\right)^2 + \lambda}$$

$$+ O(H)\cdot\frac{\left|\bar{f}_h(s,a) - \bar{f}_h'(s,a)\right|}{\sqrt{\sum_{k\in[K]}\left(\bar{f}_h(\bar{z}_h^k) - \bar{f}_h'(\bar{z}_h^k)\right)^2 + \lambda}}\cdot\sqrt{\sum_{k\in[K]}\left(\bar{f}_h(\bar{z}_h^k) - \bar{f}_h'(\bar{z}_h^k)\right)^2 + \lambda}$$

$$\le O(H)\cdot\epsilon + \frac{\left|\bar{g}_h(s,a) - \bar{g}_h'(s,a)\right|}{\sqrt{\sum_{k\in[K]}\left(\bar{g}_h(\bar{z}_h^k) - \bar{g}_h'(\bar{z}_h^k)\right)^2 + \lambda}}\cdot\sqrt{(\bar{\beta}_{2,h})^2 + \lambda}$$

$$+ O(H)\cdot\frac{\left|\bar{f}_h(s,a) - \bar{f}_h'(s,a)\right|}{\sqrt{\sum_{k\in[K]}\left(\bar{f}_h(\bar{z}_h^k) - \bar{f}_h'(\bar{z}_h^k)\right)^2 + \lambda}}\cdot\sqrt{(\bar{\beta}_{1,h})^2 + \lambda}$$

$$\le \widetilde{O}(\sqrt{\log(\mathcal{N}\cdot\mathcal{N}_b)}H^2)\cdot D_{\mathcal{F}_h}(z, \mathcal{D}_h', 1)$$

$$\le \widetilde{O}\left(\frac{\sqrt{\log(\mathcal{N}\cdot\mathcal{N}_b)}H^2}{\sqrt{K\kappa}}\right),$$

where the first inequality holds due to the completeness assumption. The second inequality holds due to (D.3) and (D.4). The third inequality holds due to Definition 3.4 . The last inequality holds due to Lemma D.1. Therefore, we have

$$\left|\mathbb{B}_h(s,a) - [\text{Var}_h \breve{f}_{h+1}](s,a)\right| \le \widetilde{O}\left(\frac{\sqrt{\log(\mathcal{N} \cdot \mathcal{N}_b)}H^2}{\sqrt{K}\kappa}\right). \tag{D.5}$$

To further bound the difference between $\left[\text{Var}_h \breve{f}_{h+1}\right](s,a)$ and $[\text{Var}_h V_{h+1}^*](s,a)$, we prove $\left\|\breve{f}_{h+1} - V_{h+1}^*\right\|_\infty \le \widetilde{O}\left(\frac{\sqrt{\log(\mathcal{N}\cdot\mathcal{N}_b)}H^3}{\sqrt{K}\kappa}\right)$ by induction.

At horizon $H+1$, $\breve{f}_{H+1} = V_{H+1}^* = 0$, the inequality holds naturally. At horizon $H$, we have

$$\begin{aligned}
Q_H^*(s,a) &= [\mathcal{T}_H V_{H+1}^*](s,a) \\
&= [\mathcal{T}_H \breve{f}_{H+1}](s,a) \\
&\ge \bar{f}_H(s,a) - \left|[\mathcal{T}_H \breve{f}_{H+1}](s,a) - \bar{f}_H(s,a)\right| \\
&\ge \bar{f}_H(s,a) - (\epsilon + |\bar{f}_H'(s,a) - \bar{f}_H(s,a)|) \\
&\ge \bar{f}_H(s,a) - \bar{b}_H(s,a) - \epsilon \\
&= \breve{f}_H(s,a),
\end{aligned}$$

where the first inequality holds due to the triangle inequality. The second inequality holds due to (D.1). The third inequality holds due to the property of the bonus function $\bar{b}_H$ and (D.3). The last equality holds due to our definition of $\breve{f}_H$ in Algorithm 1 Line 7. Therefore, $V_H^*(s) \ge \breve{f}_H(s)$ for all $s \in \mathcal{S}$.

We denote the policy derived from $\breve{f}_H$ by $\breve{\pi}_H$, i.e. $\breve{\pi}_H(s) = \text{argmax}_a \breve{f}_H(s,a)$ and then we have

$$\begin{aligned}
V_H^*(s) - \breve{f}_H(s) &= \langle Q_H^*(s,\cdot) - \breve{f}_H(s,\cdot), \pi^*(\cdot|s)\rangle_\mathcal{A} + \langle \breve{f}_H(s,\cdot), \pi_H^*(\cdot|s) - \breve{\pi}_H(\cdot|s)\rangle_\mathcal{A} \\
&\le \langle Q_H^*(s,\cdot) - \breve{f}_H(s,\cdot), \pi^*(\cdot|s)\rangle_\mathcal{A} \\
&= \langle [\mathcal{T}_H V_{H+1}^*](s,\cdot) - \bar{f}_H(s,\cdot) + \bar{b}_H(s,\cdot), \pi^*(\cdot|s)\rangle_\mathcal{A} \\
&= \langle [\mathcal{T}_H \breve{f}_{H+1}](s,\cdot) - \bar{f}_H(s,\cdot) + \bar{b}_H(s,\cdot), \pi^*(\cdot|s)\rangle_\mathcal{A} \\
&\quad + \langle [\mathcal{T}_H V_{H+1}^*](s,\cdot) - [\mathcal{T}_H \breve{f}_{H+1}](s,\cdot), \pi_H^*(\cdot|s)\rangle_\mathcal{A} \\
&\le 2\langle \bar{b}_H(s,\cdot), \pi_H^*(\cdot,s)\rangle_\mathcal{A} + \epsilon \\
&\le \widetilde{O}\left(\frac{\sqrt{\log(\mathcal{N}\cdot\mathcal{N}_b)}H}{\sqrt{K}\kappa}\right),
\end{aligned}$$

where the first inequality holds because the policy $\breve{\pi}_H$ takes the action which maximizes $\breve{f}_H$. The second inequality holds due to (D.1) and (D.3). For the last inequality, we use the choice of $b_H$ such that for all $z \in \mathcal{S} \times \mathcal{A}$ with constant $0 < C < \infty$

$$\bar{b}_H(z) \le C \cdot \left(D_{\mathcal{F}_H}(z; \mathcal{D}_H; 1) \cdot \sqrt{(\bar{\beta}_H)^2 + \lambda} + \epsilon\bar{\beta}_H\right).$$

Therefore, we have $\max_z \bar{b}_H(z) \le \widetilde{O}\left(\frac{\sqrt{\log(\mathcal{N}\cdot\mathcal{N}_b)}H}{\sqrt{K}\kappa}\right)$, which uses Lemma D.1 and $\beta_H = \widetilde{O}(\sqrt{\log(\mathcal{N}\cdot\mathcal{N}_b)}H)$.

Next, we prove by induction. Let $R_h = \widetilde{O}\left(\frac{\sqrt{\log(\mathcal{N}\cdot\mathcal{N}_b)}H}{\sqrt{K}\kappa}\right) \cdot (H - h + 1)$. We define the induction assumption as follows: Suppose with the probability of $1 - \delta_{h+1}$, the event $\mathcal{E}_{h+1} = \{0 \le V_{h+1}^*(s) - \breve{f}_{h+1}(s) \le R_{h+1}\}$ holds. Then we want to prove that with the probability of $1 - \delta_h$ ($\delta_h$ will be determined later), the event $\mathcal{E}_h = \{0 \le V_h^*(s) - \breve{f}_h(s) \le R_h\}$ holds.

Conditioned on the event $\mathcal{E}_{h+1}$, using similar argument to stage $H$, we have

$$\begin{aligned}
Q_h^*(s,a) &= [\mathcal{T}_h V_{h+1}^*](s,a) \\
&\ge [\mathcal{T}_h \breve{f}_{h+1}](s,a)
\end{aligned}$$

$$\geq \bar{f}_h(s,a) - \left|[\mathcal{T}_h \check{f}_{h+1}](s,a) - \bar{f}_h(s,a)\right|$$

$$\geq \bar{f}_h(s,a) - \left(\epsilon + |\bar{f}_h'(s,a) - \bar{f}_h(s,a)|\right)$$

$$\geq \bar{f}_h(s,a) - \bar{b}_h(s,a) - \epsilon$$

$$= \check{f}_h(s,a).$$

where the first inequality holds due to $\mathcal{E}_{h+1}$. The second inequality holds due to the triangle inequality. The third inequality holds due to (D.1). The fourth inequality holds due to the property of the bonus function $\bar{b}_h$ and (D.3). The last equality holds due to our definition of $\check{f}_h$ in Algorithm 1 Line 7. Therefore, $V_h^*(\cdot) \geq \check{f}_h(\cdot)$.

On the other hand, similar to the case at horizon $H$, we denote the policy derived from $\check{f}_h$ by $\check{\pi}_h$, i.e. $\check{\pi}_h(s) = \text{argmax}_a \check{f}_h(s,a)$. Taking a union bound over the event in Lemma D.1 and $\mathcal{E}_{h+1}$, we have with probability at least $1 - \delta_{h+1} - \delta/(2H^2)$,

$$V_h^*(s) - \check{f}_h(s) = \langle Q_h^*(s,\cdot) - \check{f}_h(s,\cdot), \pi_h^*(\cdot|s)\rangle_{\mathcal{A}} + \langle \check{f}_h(s,\cdot), \pi_h^*(\cdot|s) - \check{\pi}_h(\cdot|s)\rangle_{\mathcal{A}}$$

$$\leq \langle Q_h^*(s,\cdot) - \check{f}_h(s,\cdot), \pi_h^*(\cdot|s)\rangle_{\mathcal{A}}$$

$$= \langle [\mathcal{T}_h V_{h+1}^*](s,\cdot) - \bar{f}_h(s,\cdot) + \bar{b}_h(s,a), \pi_h^*(\cdot|s)\rangle_{\mathcal{A}} + \epsilon$$

$$= \langle [\mathcal{T}_h \check{f}_{h+1}](s,\cdot) - \bar{f}_h(s,\cdot) + \bar{b}_h(s,a), \pi_h^*(\cdot|s)\rangle_{\mathcal{A}}$$

$$\qquad + \langle [\mathcal{T}_h V_{h+1}^*](s,\cdot) - [\mathcal{T}_h \check{f}_{h+1}](s,\cdot), \pi_h^*(\cdot|s)\rangle_{\mathcal{A}} + \epsilon$$

$$\leq 2\langle \bar{b}_h(s,\cdot), \pi_h^*(\cdot,s)\rangle_{\mathcal{A}} + 2\epsilon + R_{h+1}$$

$$\leq R_{h+1} + \widetilde{O}\left(\frac{\sqrt{\log(\mathcal{N}\cdot\mathcal{N}_b)}H}{\sqrt{K\kappa}}\right)$$

$$\leq \widetilde{O}\left(\frac{\sqrt{\log(\mathcal{N}\cdot\mathcal{N}_b)}H}{\sqrt{K\kappa}}\right) \cdot (H-h+1) = R_h,$$

where the first inequality holds because the policy $\check{\pi}_H$ takes the action which maximizes $\check{f}_H$. The second inequality holds due to (D.1) and (D.3). The third inequality holds due to Lemma D.1. The last inequality holds due to the induction assumption. Therefore, we can choose $\delta_h = (H-h+1)\delta/(2H^2) \leq \delta/2H$. Taking a union bound over all $h \in [H]$, we prove that with probability at least $1 - \delta/2$, the following inequality holds for all $h \in [H]$ simultaneously

$$0 \leq V_{h+1}^*(\cdot) - \check{f}_{h+1}(\cdot) \leq \widetilde{O}\left(\frac{\sqrt{\log(\mathcal{N}\cdot\mathcal{N}_b)}H^2}{\sqrt{K\kappa}}\right). \tag{D.6}$$

Conditioned on this event, we can further bound the difference between $[\text{Var}_h \check{f}_{h+1}](s,a)$ and $[\text{Var}_h V_{h+1}^*](s,a)$.

$$\left|[\text{Var}_h \check{f}_{h+1}](s,a) - [\text{Var}_h V_{h+1}^*](s,a)\right|$$

$$\leq \left|[\mathbb{P}_h \check{f}_{h+1}^2](s,a) - [\mathbb{P}_h V_{h+1}^{*2}](s,a)\right| + \left|\left([\mathbb{P}_h \check{f}_{h+1}](s,a)\right)^2 - \left([\mathbb{P}_h V_{h+1}^*](s,a)\right)^2\right|$$

$$\leq O(H) \cdot \left\|V_{h+1}^* - \check{f}_{h+1}\right\|_{\infty}$$

$$\leq \widetilde{O}\left(\frac{\sqrt{\log(\mathcal{N}\cdot\mathcal{N}_b)}H^3}{\sqrt{K\kappa}}\right), \tag{D.7}$$

where the first inequality holds due to the triangle inequality. The second inequality holds due to $V_{h+1}^*, \check{f}_{h+1} \leq H$. The last inequality holds due to (D.6). Therefore, for any $(s,a) \in \mathcal{S} \times \mathcal{A}$, we have

$$\left|\mathbb{B}_h(s,a) - [\text{Var}_h V_{h+1}^*](s,a)\right| \leq \left|\mathbb{B}_h(s,a) - [\text{Var}_h \check{f}_{h+1}](s,a)\right|$$

$$+ \left|[\text{Var}_h \check{f}_{h+1}](s,a) - [\text{Var}_h V_{h+1}^*](s,a)\right|$$

$$\leq \widetilde{O}\left(\frac{\sqrt{\log(\mathcal{N}\cdot\mathcal{N}_b)}H^3}{\sqrt{K\kappa}}\right).$$

Thus, for any $(s,a)\in\mathcal{S}\times\mathcal{A}$, we have

$$\mathbb{B}_h(s,a) - \widetilde{O}\left(\frac{\sqrt{\log(\mathcal{N}\cdot\mathcal{N}_b)}H^3}{\sqrt{K\kappa}}\right) \leq [\mathrm{Var}_h V_{h+1}^*](s,a),$$

where the first inequality holds due to the triangle inequality. The second inequality holds due to (D.5) and (D.7). Finally, using the fact that the function $\max\{1,\cdot\}$ is increasing and nonexpansive, we complete the proof of Lemma D.4, which is

$$[\mathbb{V}_h V_{h+1}^*](s,a) - \widetilde{O}\left(\frac{\sqrt{\log(\mathcal{N}\cdot\mathcal{N}_b)}H^3}{\sqrt{K\kappa}}\right) \leq \widehat{\sigma}_h^2(s,a) \leq [\mathbb{V}_h V_{h+1}^*](s,a).$$

$\square$

# E PROOF OF LEMMAS IN SECTION D

## E.1 PROOF OF LEMMA D.1

*Proof of Lemma D.1.* From the definition of $D^2$ divergence (Definition 3.4), we have

$$D_{\mathcal{F}_h}^2(z;\mathcal{D}_h;1) = \sup_{f_1,f_2\in\mathcal{F}_h} \frac{(f_1(z)-f_2(z))^2}{\sum_{k\in[K]}\left(f_1(z_h^k)-f_2(z_h^k)\right)^2+\lambda} \tag{E.1}$$

By the Hoeffding's inequality (Lemma I.3), with probability at least $1-\delta/(\mathcal{N}^2)$, we have

$$\sum_{k\in[K]}\left(f_1(z_h^k)-f_2(z_h^k)\right)^2 - K\mathbb{E}_{\mu,h}\left[(f_1(z_h)-f_2(z_h))^2\right] \geq -2\sqrt{2K\log(\mathcal{N}^2/\delta)}\cdot\|f_1-f_2\|_\infty^2.$$

Hence, after taking a union bound, we have with probability at least $1-\delta$, for all $f_1,f_2\in\mathcal{F}_h$,

$$\sum_{k\in[K]}\left(f_1(z_h^k)-f_2(z_h^k)\right)^2 \geq K\mathbb{E}_{\mu,h}\left[(f_1(z_h)-f_2(z_h))^2\right] - 2\sqrt{2K\log(\mathcal{N}^2/\delta)}\cdot\|f_1-f_2\|_\infty^2$$

$$\geq K\cdot\kappa\|f_1-f_2\|_\infty^2 - 2\sqrt{2K\log(\mathcal{N}^2/\delta)}\cdot\|f_1-f_2\|_\infty^2, \tag{E.2}$$

where the second inequality holds due to Assumption 3.6. Substituting (E.2) into (E.1), when the size of dataset $K \geq \widetilde{\Omega}\left(\frac{\log\mathcal{N}}{\kappa^2}\right)$, we have

$$D_{\mathcal{F}_h}^2(z;\mathcal{D}_h;1) \leq \sup_{f_1,f_2\in\mathcal{F}_h} \frac{(f_1(z)-f_2(z))^2}{\frac{1}{2}K\cdot\kappa\|f_1-f_2\|_\infty^2+\lambda} = \widetilde{O}\left(\frac{1}{K\kappa}\right).$$

$\square$

## E.2 PROOF OF LEMMA D.2

In the proof of Lemma D.2, we need to prove the following concentration inequality.

**Lemma E.1.** Based on the dataset $\mathcal{D}' = \{\bar{s}_h^k, \bar{a}_h^k, \bar{r}_h^k\}_{k,h=1}^{K,H}$, we define the filtration

$$\bar{\mathcal{H}}_h^k = \sigma\left(\bar{s}_1^1, \bar{a}_1^1, \bar{r}_1^1, \bar{s}_2^1, \ldots, \bar{r}_H^1, \bar{s}_{H+1}^1; \bar{s}_1^2, \bar{a}_1^2, \bar{r}_1^2, \bar{s}_2^2, \ldots, \bar{r}_H^2, \bar{s}_{H+1}^2; \ldots; \bar{s}_1^k, \bar{a}_1^k, \bar{r}_1^k, \bar{s}_2^k, \ldots, \bar{r}_h^k, \bar{s}_{h+1}^k\right).$$

For any fixed functions $f, f' : \mathcal{S} \to [0, H]$, we make the following definitions:

$$\bar{\eta}_h^k[f'] := f'(\bar{s}_{h+1}^k) - [\mathbb{P}_h f'](\bar{s}_h^k, \bar{a}_h^k)$$

$$\bar{D}_h^k[f, f'] := 2\bar{\eta}_h^k[f']\left(f(\bar{z}_h^k) - [\mathcal{T}_h f'](\bar{z}_h^k)\right).$$

Then with probability at least $1 - \delta/(4H^2\mathcal{N}^2\mathcal{N}_b^2)$, the following inequality holds,

$$\sum_{k\in[K]} \bar{D}_h^k[f, f'] \leq (24H+5)i^2(\delta) + \frac{\sum_{k\in[K]}\left(f(\bar{z}_h^k) - [\mathcal{T}_h f'](\bar{z}_h^k)\right)^2}{2},$$

where $i(\delta) = \sqrt{2\log\frac{(\mathcal{N}\cdot\mathcal{N}_b)H(2\log(4K)+2)(\log(2L)+2)}{\delta}}$.

*Proof of Lemma D.2.* Let $(\bar{\beta}_{1,h})^2 = (48H^2 + 10)i^2(\delta) + 16KH\epsilon$. We define the event $\bar{\mathcal{E}}_{1,h} := \left\{ \sum_{k \in [K]} \left( \bar{f}'_h(\bar{z}^k_h) - \bar{f}_h(\bar{z}^k_h) \right)^2 > (\bar{\beta}_{1,h})^2 \right\}$. We have the following inequality:

$$
\sum_{k \in [K]} \left( \bar{f}'_h(\bar{z}^k_h) - \bar{f}_h(\bar{z}^k_h) \right)^2
$$

$$
= \sum_{k \in [K]} \left[ \left( \bar{r}^k_h + \check{f}_{h+1}(\bar{s}^k_{h+1}) - \bar{f}'_h(\bar{z}^k_h) \right) + \left( \bar{f}_h(\bar{z}^k_h) - \bar{r}^k_h - \check{f}_{h+1}(\bar{s}^k_{h+1}) \right) \right]^2
$$

$$
= \sum_{k \in [K]} \left( \bar{r}^k_h + \check{f}_{h+1}(\bar{s}^k_{h+1}) - \bar{f}'_h(\bar{z}^k_h) \right)^2 + \sum_{k \in [K]} \left( \bar{f}_h(\bar{z}^k_h) - \bar{r}^k_h - \check{f}_{h+1}(\bar{s}^k_{h+1}) \right)^2
$$

$$
+ 2 \sum_{k \in [K]} \left( \bar{r}^k_h + \check{f}_{h+1}(\bar{s}^k_{h+1}) - \bar{f}'_h(\bar{z}^k_h) \right) \left( \bar{f}_h(\bar{z}^k_h) - \bar{r}^k_h - \check{f}_{h+1}(\bar{s}^k_{h+1}) \right)
$$

$$
\leq 2 \sum_{k \in [K]} \left( \bar{r}^k_h + \check{f}_{h+1}(\bar{s}^k_{h+1}) - \bar{f}'_h(\bar{z}^k_h) \right)^2
$$

$$
+ 2 \sum_{k \in [K]} \left( \bar{r}^k_h + \check{f}_{h+1}(\bar{s}^k_{h+1}) - \bar{f}'_h(\bar{z}^k_h) \right) \left( \bar{f}_h(\bar{z}^k_h) - \bar{r}^k_h - \check{f}_{h+1}(\bar{s}^k_{h+1}) \right)
$$

$$
= 2 \sum_{k \in [K]} \left( \bar{r}^k_h + \check{f}_{h+1}(\bar{s}^k_{h+1}) - \bar{f}'_h(\bar{z}^k_h) \right) \left( \bar{f}_h(\bar{z}^k_h) - \bar{f}'_h(\bar{z}^k_h) \right), \tag{E.3}
$$

where the first inequality holds due to our choice of $\bar{f}_h$, i.e.,

$$
\bar{f}_h = \operatorname*{argmin}_{f_h \in \mathcal{F}_h} \sum_{k \in [K]} \left( f_h(\bar{s}^k_h, \bar{a}^k_h) - \bar{r}^k_h - \check{f}_{h+1}(\bar{s}^k_{h+1}) \right)^2.
$$

Next, we will use Lemma E.1. For any fixed $h$, let $f = \bar{f}_h \in \mathcal{F}_h$, $f' = \check{f}_{h+1} = \{\bar{f} - \epsilon\}_{[0,H-h+1]}$, where $\bar{f} = \bar{f}_h - \bar{b}_h \in \mathcal{F}_h - \mathcal{W}$. Following the construction in Lemma E.1, we define

$$
\bar{\eta}^k_h[f'] = \bar{r}^s_h + f'(\bar{s}^k_{h+1}) - \mathbb{E}\left[ \bar{r}^k_h + f'(\bar{s}^k_{h+1}) | \bar{z}^k_h \right],
$$

$$
\text{and } \bar{D}^k_h[f, f'] = 2\bar{\eta}^k_h[f'] \left( f(\bar{z}^k_h) - [\mathcal{T}_h \bar{f}'](\bar{z}^k_h) \right).
$$

Due to the result of Lemma E.1, taking a union bound on the function class, with probability at least $1 - \delta/(4H^2)$, the following inequality holds,

$$
\sum_{k \in [K]} \bar{D}^k_h[f, f'] \leq (24H^2 + 5)i^2(\delta) + \frac{\sum_{k \in [K]} \left( f(\bar{z}^k_h) - [\mathcal{T}_h f'](\bar{z}^k_h) \right)^2}{2}. \tag{E.4}
$$

Therefore, with probability at least $1 - \delta/(4H^2)$, we have

$$
2 \sum_{k \in [K]} \left( \bar{r}^k_h + \check{f}_{h+1}(\bar{s}^k_{h+1}) - \bar{f}'_h(\bar{z}^k_h) \right) \left( \bar{f}_h(\bar{z}^k_h) - \bar{f}'_h(\bar{z}^k_h) \right)
$$

$$
= 2 \sum_{k \in [K]} \left( \bar{r}^k_h + \check{f}_{h+1}(\bar{s}^k_{h+1}) - [\mathcal{T}_h \check{f}_{h+1}](\bar{z}^k_h) \right) \left( \bar{f}_h(\bar{z}^k_h) - \bar{f}'_h(\bar{z}^k_h) \right)
$$

$$
+ 2 \sum_{k \in [K]} \left( [\mathcal{T}_h \check{f}_{h+1}](\bar{z}^k_h) - \bar{f}'_h(\bar{z}^k_h) \right) \left( \bar{f}_h(\bar{z}^k_h) - \bar{f}'_h(\bar{z}^k_h) \right) \tag{E.5}
$$

$$
\leq 2 \sum_{k \in [K]} \left( \bar{r}^k_h + \check{f}_{h+1}(\bar{s}^k_{h+1}) - [\mathcal{T}_h \check{f}_{h+1}](\bar{z}^k_h) \right) \left( \bar{f}_h(\bar{z}^k_h) - \bar{f}'_h(\bar{z}^k_h) \right) + 4KH\epsilon
$$

$$
\leq (24H^2 + 5)i^2(\delta) + 4KH\epsilon + \frac{\sum_{k \in [K]} \left( \bar{f}_h(\bar{z}^k_h) - [\mathcal{T}_h \check{f}_{h+1}](\bar{z}^k_h) \right)^2}{2} \tag{E.6}
$$

$$
\leq (24H^2 + 5)i^2(\delta) + 8KL\epsilon + \frac{\sum_{k \in [K]} \left( \bar{f}'_h(\bar{z}^k_h) - \bar{f}_h(\bar{z}^k_h) \right)^2}{2}
$$

$$= \frac{(\bar{\beta}_{1,h})^2}{2} + \frac{\sum_{k\in[K]} \left( \bar{f}'_h(\bar{z}^k_h) - \widetilde{f}'_h(\bar{z}^k_h) \right)^2}{2}. \tag{E.7}$$

where the first and third inequalities hold because of the completeness assumption. The second inequality holds due to (E.4). The last equality holds due to the choice of

$$\bar{\beta}_{1,h} = \sqrt{2(24L^2+5)i^2(\delta) + 16KL\epsilon} = \widetilde{O}\left( \sqrt{\log(\mathcal{N}\cdot\mathcal{N}_b)}H \right).$$

However, conditioned on the event $\bar{\mathcal{E}}_{1,h}$, we have

$$2 \sum_{k\in[K]} \left( \bar{r}^k_h + \breve{f}_{h+1}(\bar{s}^k_{h+1}) - \bar{f}'_h(\bar{z}^k_h) \right) \left( \bar{f}_h(\bar{z}^k_h) - \bar{f}'_h(\bar{z}^k_h) \right)$$

$$\geq \sum_{k\in[K]} \left( \bar{f}'_h(\bar{z}^k_h) - \bar{f}_h(\bar{z}^k_h) \right)^2$$

$$> \frac{(\bar{\beta}_{1,h})^2}{2} + \frac{\sum_{k\in[K]} \left( \bar{f}'_h(\bar{z}^k_h) - \bar{f}_h(\bar{z}^k_h) \right)^2}{2}.$$

where the first inequality holds due to (E.3). The second inequality holds due to $\bar{\mathcal{E}}_{1,h}$. This is contradictory with (E.7). Thus, we have $\mathbb{P}[\bar{\mathcal{E}}_{1,h}] \leq \delta/(4H^2)$ and complete the proof of Lemma D.2. $\qquad\square$

## E.3 PROOF OF LEMMA D.3
To prove this lemma, we need a lemma similar to Lemma E.1

**Lemma E.2.** On dataset $\mathcal{D}' = \{\bar{s}^k_h, \bar{a}^k_h, \bar{r}^k_h\}^{K,H}_{k,h=1}$, we define the filtration

$$\bar{\mathcal{H}}^k_h = \sigma(\bar{s}^1_1, \bar{a}^1_1, \bar{r}^1_1, \bar{s}^1_2, \ldots, \bar{r}^1_H, \bar{s}^1_{H+1}; \bar{s}^2_1, \bar{a}^2_1, \bar{r}^2_1, \bar{s}^2_2, \ldots, \bar{r}^2_H, \bar{s}^2_{H+1}; \ldots; \bar{s}^k_1, \bar{a}^k_1, \bar{r}^k_1, \bar{s}^k_2, \ldots, \bar{r}^k_h, \bar{s}^k_{h+1}).$$

For any fixed function $f, f' : \mathcal{S} \to [0, H]$, we make the following definitions:

$$\bar{\eta}^k_h[f'] := \left( \bar{r}^k_h + f'(\bar{s}^k_{h+1}) \right)^2 - \left[ \mathbb{P}_h(\bar{r}_h + f')^2 \right] \left( \bar{s}^k_h, \bar{a}^k_h \right)$$
$$\bar{D}^k_h[f, f'] := 2\bar{\eta}^k_h[f'] \left( f(\bar{z}^k_h) - [\mathcal{T}_{2,h}f'](\bar{z}^k_h) \right).$$

Then with probability at least $1 - \delta/(4H^2\mathcal{N}^2\mathcal{N}^2_b)$, the following inequality holds,

$$\sum_{k\in[K]} \bar{D}^k_h[f, f'] \leq (24H+5)i'^2(\delta) + \frac{\sum_{k\in[K]} \left( f(\bar{z}^k_h) - [\mathcal{T}_{2,h}f'](\bar{z}^k_h) \right)^2}{2},$$

where $i'(\delta) = \sqrt{4\log \frac{(\mathcal{N}\cdot\mathcal{N}_b)H(2\log(4LK)+2)(\log(4L)+2)}{\delta}}$.

*Proof of Lemma D.3.* Let $(\bar{\beta}_{2,h})^2 = (40H^4+10)i'^2(\delta) + 16KL\epsilon$. We define the event $\bar{\mathcal{E}}_{2,h} := \left\{ \sum_{k\in[K]} \left( \bar{g}'_h(\bar{z}^k_h) - \bar{g}_h(\bar{z}^k_h) \right)^2 > (\bar{\beta}_{2,h})^2 \right\}$. We can prove the following inequality:

$$\sum_{k\in[K]} \left( \bar{g}'_h(\bar{z}^k_h) - \bar{g}_h(\bar{z}^k_h) \right)^2$$

$$= \sum_{k\in[K]} \left[ \left( \left( \bar{r}^k_h + \breve{f}_{h+1}(\bar{s}^k_{h+1}) \right)^2 - \bar{g}'_h(\bar{z}^k_h) \right) + \left( \bar{g}_h(\bar{z}^k_h) - \left( \bar{r}^k_h + \breve{f}_{h+1}(\bar{s}^k_{h+1}) \right)^2 \right) \right]^2$$

$$= \sum_{k\in[K]} \left( \left( \bar{r}^k_h + \breve{f}_{h+1}(\bar{s}^k_{h+1}) \right)^2 - \bar{g}'_h(\bar{z}^k_h) \right)^2 + \sum_{k\in[K]} \left( \bar{g}_h(\bar{z}^k_h) - \left( \bar{r}^k_h + \breve{f}_{h+1}(\bar{s}^k_{h+1}) \right)^2 \right)^2$$

$$+ 2 \sum_{k\in[K]} \left( \left( \bar{r}^k_h + \breve{f}_{h+1}(\bar{s}^k_{h+1}) \right)^2 - \bar{g}'_h(\bar{z}^k_h) \right) \left( \bar{g}_h(\bar{z}^k_h) - \left( \bar{r}^k_h + \breve{f}_{h+1}(\bar{s}^k_{h+1}) \right)^2 \right)$$

$$\leq 2 \sum_{k\in[K]} \left( \left( \bar{r}^k_h + \breve{f}_{h+1}(\bar{s}^k_{h+1}) \right)^2 - \bar{g}'_h(\bar{z}^k_h) \right)^2$$

$$+ 2 \sum_{k \in [K]} \left( \left( \bar{r}_h^k + \breve{f}_{h+1}(\bar{s}_{h+1}^k) \right)^2 - \bar{g}_h'(\bar{z}_h^k) \right) \left( \bar{g}_h(\bar{z}_h^k) - \left( \bar{r}_h^k + \breve{f}_{h+1}(\bar{s}_{h+1}^k) \right)^2 \right)$$

$$= 2 \sum_{k \in [K]} \left( \left( \bar{r}_h^k + \breve{f}_{h+1}(\bar{s}_{h+1}^k) \right)^2 - \bar{g}_h'(\bar{z}_h^k) \right) \left( \bar{g}_h(\bar{z}_h^k) - \bar{g}_h'(\bar{z}_h^k) \right), \tag{E.8}$$

where the first inequality holds due to our choice of $\bar{g}_h$, i.e.

$$\bar{g}_h = \operatorname*{argmin}_{g_h \in \mathcal{F}_h} \sum_{k \in [K]} \left( g_h(\bar{s}_h^k, \bar{a}_h^k) - (\bar{r}_h^k + \breve{f}_{h+1}(\bar{s}_{h+1}^k))^2 \right)^2.$$

Next, we will use Lemma E.2. For any fixed $h$, let $f = \bar{g}_h \in \mathcal{F}_h$, $f' = \breve{f}_{h+1} = \{\bar{f} - \epsilon\}_{[0, H-h+1]}$, where $\bar{f} = \bar{f}_h - \bar{b}_h \in \mathcal{F}_h - \mathcal{W}$. Following the construction in Lemma E.2, we define

$$\bar{\eta}_h^k[f'] := \left( \bar{r}_h^k + f'(\bar{s}_{h+1}^k) \right)^2 - \left[ \mathbb{P}_h(\bar{r}_h + f')^2 \right] (\bar{s}_h^k, \bar{a}_h^k)$$
$$\text{and } \bar{D}_h^k[f, f'] := 2\bar{\eta}_h^k[f'] \left( f(\bar{z}_h^k) - [\mathcal{T}_{2,h}f'](\bar{z}_h^k) \right).$$

Due to the result of Lemma E.2, taking a union bound on the function, with probability at least $1 - \delta/(4H^2)$, the following inequality holds,

$$\sum_{k \in [K]} \bar{D}_h^k[f, f'] \le (20H^4 + 5)i^2(\delta) + \frac{\sum_{k \in [K]} \left( f(\bar{z}_h^k) - [\mathcal{T}_{2,h}f'](\bar{z}_h^k) \right)^2}{2}. \tag{E.9}$$

Therefore, with probability at least $1 - \delta/(4H^2)$, we have

$$2 \sum_{k \in [K]} \left( \left( \bar{r}_h^k + \breve{f}_{h+1}(\bar{s}_{h+1}^k) \right)^2 - \bar{g}_h'(\bar{z}_h^k) \right) \left( \bar{g}_h(\bar{z}_h^k) - \bar{g}_h'(\bar{z}_h^k) \right)$$

$$= 2 \sum_{k \in [K]} \left( \left( \bar{r}_h^k + \breve{f}_{h+1}(\bar{s}_{h+1}^k) \right)^2 - [\mathcal{T}_{2,h}\breve{f}_{h+1}](\bar{z}_h^k) \right) \left( \bar{g}_h(\bar{z}_h^k) - \bar{g}_h'(\bar{z}_h^k) \right)$$

$$\quad + 2 \sum_{k \in [K]} \left( [\mathcal{T}_{2,h}\breve{f}_{h+1}](\bar{z}_h^k) - \bar{g}_h'(\bar{z}_h^k) \right) \left( \bar{g}_h(\bar{z}_h^k) - \bar{g}_h'(\bar{z}_h^k) \right)$$

$$\le 2 \sum_{k \in [K]} \left( \left( \bar{r}_h^k + \breve{f}_{h+1}(\bar{s}_{h+1}^k) \right)^2 - [\mathcal{T}_{2,h}\breve{f}_{h+1}](\bar{z}_h^k) \right) \left( \bar{g}_h(\bar{z}_h^k) - \bar{g}_h'(\bar{z}_h^k) \right) + 4KL\epsilon$$

$$\le (20H^4 + 5)i'^2(\delta) + 4KL\epsilon + \frac{\sum_{k \in [K]} \left( \bar{g}_h(\bar{z}_h^k) - [\mathcal{T}_{2,h}\breve{f}_{h+1}](\bar{z}_h^k) \right)^2}{2}$$

$$\le (20H^4 + 5)i'^2(\delta) + 8KL\epsilon + \frac{\sum_{k \in [K]} \left( \bar{g}_h'(\bar{z}_h^k) - \bar{g}_h(\bar{z}_h^k) \right)^2}{2}$$

$$\le \frac{(\bar{\beta}_{2,h})^2}{2} + \frac{\sum_{k \in [K]} \left( \bar{g}_h'(\bar{z}_h^k) - \bar{g}_h(\bar{z}_h^k) \right)^2}{2}, \tag{E.10}$$

where the first and third inequalities hold due to the Bellman completeness assumption. The second inequality holds due to (E.9). The last inequality holds due to the choice of

$$\bar{\beta}_{2,h} = \sqrt{(40H^4 + 10)i'^2(\delta) + 16KL\epsilon} = \widetilde{O}(\sqrt{\log(\mathcal{N} \cdot \mathcal{N}_b)}H^2).$$

However, conditioned on the event $\bar{\mathcal{E}}_{2,h}$, we have

$$2 \sum_{k \in [K]} \left( (\bar{r}_h^k + \breve{f}_{h+1}(\bar{s}_{h+1}^k))^2 - \bar{g}_h'(\bar{z}_h^k) \right) \left( \bar{g}_h(\bar{z}_h^k) - \bar{g}_h'(\bar{z}_h^k) \right)$$

$$\ge \sum_{k \in [K]} \left( \bar{g}_h'(\bar{z}_h^k) - \bar{g}_h(\bar{z}_h^k) \right)^2$$

$$> \frac{(\bar{\beta}_{2,h})^2}{2} + \frac{\sum_{k \in [K]} \left( \bar{g}_h'(\bar{z}_h^k) - \bar{g}_h(\bar{z}_h^k) \right)^2}{2},$$

where the first inequality holds due to (E.8). The last inequality holds due to $\bar{\mathcal{E}}_{2,h}$. It is contradictory with (E.10). Thus, we have $\mathbb{P}[\bar{\mathcal{E}}_{2,h}] \le \delta/(4H^2)$ and complete the proof of Lemma D.3. $\qquad \square$

## F PROOF OF THEOREM 5.1

In this section, we prove Theorem 5.1. The proof idea is similar to that of Section D. To start with, we prove that our data coverage assumption (Assumption 3.6) can lead to an upper bound of the weighted $D^2$-divergence for a large dataset.

**Lemma F.1.** Let $\mathcal{D}_h$ be a dataset satisfying Assumption 3.6. When the size of data set satisfies $K \geq \widetilde{\Omega}\left(\frac{\log \mathcal{N}}{\kappa^2}\right)$, $\widehat{\sigma}_h \leq H$, with probability at least $1 - \delta$, for each state-action pair $z$, we have

$$D_{\mathcal{F}_h}(z, \mathcal{D}_h, \widehat{\sigma}_h^2) = \widetilde{O}\left(\frac{H}{\sqrt{K\kappa}}\right).$$

With Lemma D.4, we can prove a variance-weighted version of concentration inequality.

**Lemma F.2** (Restatement of Lemma 6.3). Suppose the variance function $\widehat{\sigma}_h$ satisfies the inequality in Lemma D.4. at stage $h \in [H]$, the estimated value function $\widehat{f}_{h+1}$ in Algorithm 1 is bounded by $H$. According to Assumption 3.3, there exists some function $\bar{f}_h \in \mathcal{F}_h$, such that $|\bar{f}_h(z_h) - [\mathcal{T}_h \widehat{f}_{h+1}](z)| \leq \epsilon$ for all $z_h = (s_h, a_h)$. Then with probability at least $1 - \delta/2$, the following inequality holds for all stage $h \in [H]$ simultaneously,

$$\sum_{k \in [K]} \frac{1}{(\widehat{\sigma}_h(z_h^k))^2} \left(\bar{f}_h(z_h^k) - \widetilde{f}_h(z_h^k)\right)^2 \leq (\beta_h)^2.$$

With these lemmas, we can start the proof of Theorem 5.1.

*Proof of Theorem 5.1.* For any state-action pair $z = (s, a) \in \mathcal{S} \times \mathcal{A}$, we have

$$\left|[\mathcal{T}_h \widehat{f}_{h+1}](z) - \widetilde{f}_h(z)\right| \leq \underbrace{\left|[\mathcal{T}_h \widehat{f}_{h+1}](z) - \bar{f}_h(z)\right|}_{I_1} + \left|\bar{f}_h(z) - \widetilde{f}_h(z)\right|$$

$$\leq \epsilon + b_h(z),$$

where we bound $I_1$ with the Bellman completeness assumption. For the second term, we use the property of the bonus function and Lemma F.2. Using Lemma I.2, we have

$$V_1^*(s) - V_1^{\widehat{\pi}}(s) \leq 2 \sum_{h=1}^{H} \mathbb{E}_{\pi^*}\left[b_h(s_h, a_h) \mid s_1 = s\right] + 2\epsilon H$$

$$\leq \sum_{h=1}^{H} \mathbb{E}_{\pi^*}\left[D_{\mathcal{F}_h}(z_h; \mathcal{D}_h; \widehat{\sigma}_h^2) \cdot \sqrt{(\beta_h)^2 + \lambda} \mid s_1 = s\right] + 2\epsilon H$$

$$\leq \widetilde{O}\left(\sqrt{\log \mathcal{N}}\right) \sum_{h=1}^{H} \mathbb{E}_{\pi^*}\left[D_{\mathcal{F}_h}(z_h; \mathcal{D}_h; \widehat{\sigma}_h^2) \mid s_1 = s\right]$$

$$\leq \widetilde{O}\left(\sqrt{\log \mathcal{N}}\right) \sum_{h=1}^{H} \mathbb{E}_{\pi^*}\left[D_{\mathcal{F}_h}(z_h; \mathcal{D}_h; [\mathbb{V}_h V_{h+1}^*](\cdot, \cdot)) \mid s_1 = s\right],$$

where the first inequality holds due to Lemma I.2. The second inequality holds due to the property of the bonus function

$$b_h(z) \leq C \cdot \left(D_{\mathcal{F}_h}(z; \mathcal{D}_h; \widehat{\sigma}_h^2) \cdot \sqrt{(\beta_h)^2 + \lambda} + \epsilon \beta_h\right).$$

The third inequality holds due to our choice of $\beta_h = \widetilde{O}\left(\sqrt{\log \mathcal{N}}\right)$. The last inequality holds due to Lemma D.4 and the fact that $D^2$-divergence is increasing with respect to the variance function. We complete the proof of Theorem 5.1. $\qquad\square$

## G PROOF OF THE LEMMAS IN SECTION F

### G.1 PROOF OF LEMMA F.1

*Proof of Lemma F.1.* From the definition of $D^2$ divergence, we have

$$D_{\mathcal{F}_h}^2(z; \mathcal{D}_h; \widehat{\sigma}_h^2) = \sup_{f_1, f_2 \in \mathcal{F}_h} \frac{(f_1(z) - f_2(z))^2}{\sum_{k \in [K]} \frac{1}{(\widehat{\sigma}_h(z_h^k))^2} \left(f_1(z_h^k) - f_2(z_h^k)\right)^2 + \lambda} \qquad \text{(G.1)}$$

By the Hoeffding's inequality (Lemma I.3), with probability at least $1 - \delta/(\mathcal{N}^2)$,

$$\sum_{k \in [K]} \left( f_1(z_h^k) - f_2(z_h^k) \right)^2 - K \mathbb{E}_{\mu,h} \left[ (f_1(z_h) - f_2(z_h))^2 \right] \geq -2\sqrt{2K \log(\mathcal{N}^2/\delta)} \cdot \|f_1 - f_2\|_\infty^2.$$

Hence, after taking a union bound, we have with probability at least $1 - \delta$, for all $f_1, f_2 \in \mathcal{F}_h$,

$$\sum_{k \in [K]} \frac{1}{(\widehat{\sigma}_h(z_h^k))^2} \left( f_1(z_h^k) - f_2(z_h^k) \right)^2$$

$$\geq \frac{1}{H^2} \left( K \mathbb{E}_{\mu,h} \left[ (f_1(z_h) - f_2(z_h))^2 \right] - 2\sqrt{2K \log(\mathcal{N}^2/\delta)} \cdot \|f_1 - f_2\|_\infty^2 \right)$$

$$\geq \frac{1}{H^2} \left( K \cdot \kappa \|f_1 - f_2\|_\infty^2 - 2\sqrt{2K \log(\mathcal{N}^2/\delta)} \cdot \|f_1 - f_2\|_\infty^2 \right), \tag{G.2}$$

where the last inequality holds due to Assumption 3.6. Substituting (G.2) into (G.1), when $K \geq \widetilde{\Omega}\left( \frac{\log \mathcal{N}}{\kappa} \right)$, we have

$$D_{\mathcal{F}_h}^2(z; \mathcal{D}_h; \widehat{\sigma}_h^2) \leq \sup_{f_1, f_2 \in \mathcal{F}_h} \frac{H^2 (f_1(z) - f_2(z))^2}{\frac{1}{2} K \cdot \kappa \|f_1 - f_2\|_\infty^2 + \lambda} = \widetilde{O}\left( \frac{H^2}{K\kappa^2} \right).$$

$\square$

## G.2 PROOF OF LEMMA F.2

In this section, we assume the high probability event in Lemma D.4 holds, i.e., the following inequality holds:

$$[\mathbb{V}_h V_{h+1}^*](s,a) - \widetilde{O}\left( \frac{\sqrt{\log(\mathcal{N} \cdot \mathcal{N}_b)} H^3}{\sqrt{K\kappa}} \right) \leq \widehat{\sigma}_h^2(s,a) \leq [\mathbb{V}_h V_{h+1}^*](s,a). \tag{G.3}$$

To prove Lemma F.2, we need the following lemmas.

**Lemma G.1.** Based on the dataset $\mathcal{D} = \{s_h^k, a_h^k, r_h^k\}_{k,h=1}^{K,H}$, we define the filtration $\mathcal{H}_h^k = \sigma(s_1^1, a_1^1, r_1^1, s_2^1, \ldots, r_H^1, s_{H+1}^1; x_1^2, a_1^2, r_1^2, s_2^2, \ldots, r_H^2, s_{H+1}^2; \cdots s_1^k, a_1^k, r_1^k, s_2^k, \ldots, r_h^k, s_{h+1}^k)$. For any fixed function $f, f' : \mathcal{S} \to \in [0, L]$, we define the following random variables:

$$\eta_h^k := V_{h+1}^*(s_{h+1}^k) - [\mathbb{P}_h V_{h+1}^*](s_h^k, a_h^k)$$

$$D_h^s[f, f'] := 2 \frac{\eta_h^k}{(\widehat{\sigma}_h(z_h^k))^2} \left( f(z_h^k) - f'(z_h^k) \right),$$

As the variance function $\widehat{\sigma}_h$ satisfies (G.3), with probability at least $1 - \delta/(4H^2\mathcal{N}^2)$, the following inequality holds,

$$\sum_{k \in [K]} D_h^k[f, f'] \leq \frac{4}{3} \upsilon(\delta)\sqrt{\lambda} + \sqrt{2}\upsilon(\delta) + 30\upsilon^2(\delta)$$

$$+ \frac{\sum_{k \in [K]} \frac{1}{(\widehat{\sigma}_h(z_h^k))^2} \left( f(z_h^k) - f'(z_h^k) \right)^2}{4},$$

where $\upsilon(\delta) = \sqrt{2 \log \frac{H \mathcal{N}(2 \log(18LT)+2)(\log(18L)+2)}{\delta}}$.

**Lemma G.2.** Based on the dataset $\mathcal{D} = \{s_h^k, a_h^k, r_h^k\}_{k,h=1}^{K,H}$, we define the following filtration $\mathcal{H}_h^k = \sigma\left( s_1^1, a_1^1, r_1^1, s_2^1, \ldots, r_H^1, s_{H+1}^1; x_1^2, a_1^2, r_1^2, s_2^2, \ldots, r_H^2, s_{H+1}^2; \cdots s_1^k, a_1^k, r_1^k, s_2^k, \ldots, r_h^k, s_{h+1}^k \right)$. For any fixed functions $f, \widetilde{f} : \mathcal{S} \to [0, L]$ and $f' : \mathcal{S} \to [0, H]$, we define the following random variables

$$\xi_h^k[f'] := f'(s_{h+1}^k) - V_{h+1}^*(s_{h+1}^k) - \left[ \mathbb{P}_h(f' - V_{h+1}^*) \right](s_h^k, a_h^k),$$

$$\Delta_h^k \left[ f, \widetilde{f}, f' \right] := 2 \frac{\xi_h^k[f']}{(\widehat{\sigma}_h(z_h^k))^2} \left( f(z_h^k) - \widetilde{f}(z_h^k) \right),$$

As the variance function $\widehat{\sigma}_h$ satisfies (G.3), with probability at least $1 - \delta/(4H^2\mathcal{N}^3\mathcal{N}_b)$, the following inequality holds,

$$\sum_{k\in[K]} \Delta_h^k[f, \widetilde{f}, f'] \leq \left(\frac{4}{3}\iota(\delta)\sqrt{\lambda} + \sqrt{2}\iota(\delta)\right)\|f' - V_{h+1}^*\|_\infty^2 + \frac{2}{3}\iota^2(\delta)/\log\mathcal{N}_b$$

$$+ 30\iota^2(\delta)\|f' - V_{h+1}^*\|_\infty^2 + \frac{\sum_{k\in[K]}\frac{1}{(\widehat{\sigma}_h(z_h^k))^2}(f(z_h^k) - f'(z_h^k))^2}{4}.$$

where $\iota(\delta) = \sqrt{3\log\frac{H(\mathcal{N}\cdot\mathcal{N}_b)(2\log(18LT)+2)(\log(18L)+2)}{\delta}}$.

With these lemmas, we can start the proof of Lemma F.2.

*Proof of Lemma F.2.* We define the event $\mathcal{E}_h := \left\{\sum_{k\in[K]}\frac{1}{(\widehat{\sigma}(z_h^k))^2}\left(\bar{f}_h(z_h^k) - \widetilde{f}_h(z_h^k)\right)^2 > (\beta_h)^2\right\}$.
We have the following inequality:

$$\sum_{k\in[K]}\frac{1}{(\widehat{\sigma}(z_h^k))^2}\left(\bar{f}_h(z_h^k) - \widetilde{f}_h(z_h^k)\right)^2$$

$$= \sum_{k\in[K]}\frac{1}{(\widehat{\sigma}(z_h^k))^2}\left[\left(r_h^k + \widehat{f}_{h+1}(s_{h+1}^k) - \bar{f}_h(z_h^k)\right) + \left(\widetilde{f}_h(z_h^k) - r_h^k - \widehat{f}_{h+1}(s_{h+1}^k)\right)\right]^2$$

$$= \sum_{k\in[K]}\frac{1}{(\widehat{\sigma}(z_h^k))^2}\left(r_h^k + \widehat{f}_{h+1}(s_{h+1}^k) - \bar{f}_h(z_h^k)\right)^2 + \sum_{k\in[K]}\frac{1}{(\widehat{\sigma}(z_h^k))^2}\left(\widetilde{f}_h(z_h^k) - r_h^k - \widehat{f}_{h+1}(s_{h+1}^k)\right)^2$$

$$+ 2\sum_{k\in[K]}\frac{1}{(\widehat{\sigma}(z_h^k))^2}\left(r_h^k + \widehat{f}_{h+1}(s_{h+1}^k) - \bar{f}_h(z_h^k)\right)\left(\widetilde{f}_h(z_h^k) - r_h^k - \widehat{f}_{h+1}(s_{h+1}^k)\right)$$

$$\leq 2\sum_{k\in[K]}\frac{1}{(\widehat{\sigma}(z_h^k))^2}\left(r_h^k + \widehat{f}_{h+1}(s_{h+1}^k) - \bar{f}_h(z_h^k)\right)^2$$

$$+ 2\sum_{k\in[K]}\frac{1}{(\widehat{\sigma}(z_h^k))^2}\left(r_h^k + \widehat{f}_{h+1}(s_{h+1}^k) - \bar{f}_h(z_h^k)\right)\left(\widetilde{f}_h(z_h^k) - r_h^k - \widehat{f}_{h+1}(s_{h+1}^k)\right)$$

$$= 2\sum_{k\in[K]}\frac{1}{(\widehat{\sigma}(z_h^k))^2}\left(r_h^k + \widehat{f}_{h+1}(s_{h+1}^k) - \bar{f}_h(z_h^k)\right)\left(\widetilde{f}_h(z_h^k) - \bar{f}_h(z_h^k)\right). \tag{G.4}$$

where the first inequality holds due to our choice of $\widetilde{f}_h$ in Algorithm 1 Line 12,

$$\widetilde{f}_h = \underset{f_h\in\mathcal{F}_h}{\operatorname{argmin}}\sum_{k\in[K]}\frac{1}{(\widehat{\sigma}(z_h^k))^2}\left(f_h(s_h^k, a_h^k) - r_h^k - \widehat{f}_{h+1}(s_{h+1}^k)\right)^2.$$

We prove this lemma by induction. At horizon $H$, we first use Lemma G.1. Let $f = \widetilde{f}_H \in \mathcal{F}_H$, $f' = \bar{f}_H \in \mathcal{F}_H$. We define

$$\eta_H^k := V_{H+1}^*(s_{H+1}^k) - [\mathbb{P}_H V_{H+1}^*](z_H^k)$$

$$D_H^k[f, f'] := 2\frac{\eta_H^k}{(\widehat{\sigma}_H(z_H^k))^2}\left(f(z_H^k) - f'(z_H^k)\right).$$

Taking a union bound over the function class, with probability at least $1 - \delta/(4H^2)$, the following inequality holds,

$$\sum_{k\in[K]}2\frac{\eta_H^k}{\left(\widehat{\sigma}_H\left(z_H^k\right)\right)^2}\left(\widetilde{f}(z_H^k) - \bar{f}(z_H^k)\right) \leq \frac{4}{3}v(\delta)\sqrt{\lambda} + \sqrt{2}v(\delta) + 30v^2(\delta)$$

$$+ \frac{\sum_{k\in[K]}\frac{1}{\left(\widehat{\sigma}_H(z_H^k)\right)^2}(\widetilde{f}(z_H^k) - \bar{f}(z_H^k))^2}{4}. \tag{G.5}$$

Then we use Lemma G.2. Let $f = \widetilde{f}_H \in \mathcal{F}_H$, $\widetilde{f} = \bar{f}_H \in \mathcal{F}_H$, $f' = \widehat{f}_{H+1} = 0$. We define:

$$\xi_H^k[f'] := f'(s_{H+1}^k) - V_{H+1}^*(s_{H+1}^k) - [\mathbb{P}_H(f' - V_{H+1}^*)](z_H^k)$$

$$\Delta_H^k[f, \widetilde{f}, f'] := 2\frac{\xi_H^k[f']}{(\widehat{\sigma}_H(z_H^k))^2}\left(f(z_H^k) - \widetilde{f}(z_H^k)\right).$$

Taking a union bound over the function class, with probability at least $1 - \delta/(4H^2)$, we have

$$\sum_{k\in[K]} 2\frac{\xi_H^k[\widehat{f}_{H+1}]}{(\widehat{\sigma}_H(z_H^k))^2}\left(\widetilde{f}_H(z_H^k) - \bar{f}_H(z_H^k)\right) \leq \left(\frac{4}{3}\iota(\delta)\sqrt{\lambda} + \sqrt{2}\iota(\delta)\right)\|f' - V_{H+1}^*\|_\infty^2$$

$$+ \frac{2}{3}\iota^2(\delta)/\sqrt{\log\mathcal{N}_b} + 30\iota^2(\delta)\|\widehat{f}_{H+1} - V_{H+1}^*\|_\infty^2 + \frac{\sum_{k\in[K]}\frac{1}{(\widehat{\sigma}_H(z_H^k))^2}\left(\widetilde{f}_H(z_H^k) - \bar{f}_H(z_H^k)\right)^2}{4}. \tag{G.6}$$

Combining (G.5) and (G.6), we have with probability at least $1 - \delta/(2H^2)$, the following inequality holds:

$$2\sum_{k\in[K]} \frac{1}{(\widehat{\sigma}_H(z_H^k))^2}\left(r_H^k + \widehat{f}_{H+1}(s_{H+1}^k) - \bar{f}_H(z_H^k)\right)\left(\widetilde{f}_H(z_H^k) - \bar{f}_H(z_H^k)\right)$$

$$= 2\sum_{k\in[K]} \frac{1}{(\widehat{\sigma}_H(z_H^k))^2}\left(r_H^k + \widehat{f}_{H+1}(s_{H+1}^k) - [\mathcal{T}_H\widehat{f}_{H+1}](z_H^k)\right)\left(\widetilde{f}_H(z_H^k) - \bar{f}_H(z_H^k)\right)$$

$$+ 2\sum_{k\in[K]} \frac{1}{(\widehat{\sigma}_H(z_H^k))^2}\left([\mathcal{T}_H\widehat{f}_{H+1}](z_H^k) - \bar{f}_H(z_H^k)\right)\left(\widetilde{f}_H(z_H^k) - \bar{f}_H(z_H^k)\right)$$

$$\leq 2\sum_{k\in[K]} \frac{1}{(\widehat{\sigma}_H(z_H^k))^2}\left(r_H^k + \widehat{f}_{H+1}(s_{H+1}^k) - [\mathcal{T}_H\widehat{f}_{H+1}](z_H^k)\right)\left(\widetilde{f}_H(z_H^k) - \bar{f}_H(z_H^k)\right) + 4KL\epsilon$$

$$\leq \frac{4}{3}v(\delta)\sqrt{\lambda} + \sqrt{2}v(\delta) + 30v^2(\delta) + \left(\frac{4}{3}\iota(\delta)\sqrt{\lambda} + \sqrt{2}\iota(\delta)\right)\|f' - V_{H+1}^*\|_\infty^2 + \frac{2}{3}\iota^2(\delta)/\log\mathcal{N}_b$$

$$+ 30\iota^2(\delta)\|\widehat{f}_{H+1} - V_{H+1}^*\|_\infty^2 + 8KL\epsilon + \frac{\sum_{k\in[K]}\frac{1}{(\widehat{\sigma}_H(z_H^k))^2}\left(\bar{f}_H(z_H^k) - \widetilde{f}_H(z_H^k)\right)^2}{2}$$

$$\leq \frac{(\beta_H)^2}{2} + \frac{\sum_{k\in[K]}\frac{1}{(\widehat{\sigma}_H(z_H^k))^2}\left(\bar{f}_H(z_H^k) - \widetilde{f}_H(z_H^k)\right)^2}{2}, \tag{G.7}$$

where the first inequality holds due to the complete assumption. The second inequality holds due to (G.5) and (G.6). The last inequality holds due to the fact $\widehat{f}_{H+1} = V_{H+1}^* = 0$ and our choice of $\beta_H$, i.e.,

$$\beta_H = \sqrt{2\left(\frac{4}{3}v(\delta)\sqrt{\lambda} + \sqrt{2}v(\delta) + 30v^2(\delta) + \frac{2}{3}\iota^2(\delta)/\log\mathcal{N}_b + 8KL\epsilon\right)}$$

$$= \widetilde{O}(\sqrt{\log\mathcal{N}}).$$

However, conditioned on the event $\mathcal{E}_H$, we have

$$\sum_{k\in[K]} \frac{1}{(\widehat{\sigma}_H(z_H^k))^2}\left(r_H^k + \widehat{f}_{H+1}(s_{H+1}^k) - \bar{f}_H(z_H^k)\right)\left(\widetilde{f}_H(z_H^k) - \bar{f}_H(z_H^k)\right)$$

$$\geq \sum_{k\in[K]} \frac{1}{(\widehat{\sigma}_H(z_H^k))^2}\left(\bar{f}_H(z_H^k) - \widetilde{f}_H(z_H^k)\right)^2$$

$$> \frac{(\beta_H)^2}{2} + \frac{\sum_{k\in[K]}\frac{1}{(\widehat{\sigma}(z_H^k))^2}\left(\bar{f}_H(z_H^k) - \widetilde{f}_H(z_H^k)\right)^2}{2},$$

where the first inequality holds due to (G.4). The second inequality holds due to $\mathcal{E}_H$. It contradicts with (G.7). Therefore, we prove that $\mathbb{P}[\mathcal{E}_H] \geq 1 - \delta/2H^2$.

Suppose the event $\mathcal{E}_H$ holds, we can prove the following result.

$$
\begin{aligned}
Q_H^*(s,a) &= [\mathcal{T}_H V_{H+1}^*](s,a) \\
&= [\mathcal{T}_H \widehat{f}_{H+1}](s,a) \\
&\geq \widetilde{f}_H(s,a) - \left| [\mathcal{T}_H \widehat{f}_{H+1}](s,a) - \widetilde{f}_H(s,a) \right| \\
&\geq \widetilde{f}_H(s,a) - \left( \epsilon + |\bar{f}_H(s,a) - \widetilde{f}_H(s,a)| \right) \\
&\geq \widetilde{f}_H(s,a) - b_H(s,a) - \epsilon \\
&= \widehat{f}_H(s,a).
\end{aligned}
$$

where the first inequality holds due to the triangle inequality. The second inequality holds due to the completeness assumption. The third inequality holds due to the property of the bonus function and

$$
\sum_{k\in[K]} \frac{1}{(\widehat{\sigma}_h(z_h^k))^2} \left( \bar{f}_h(z_h^k) - \widetilde{f}_h(z_h^k) \right)^2 \leq (\beta_h)^2.
$$

by Lemma F.2. Therefore, $V_H^*(s) \geq \widehat{f}_H(s)$ for all $s \in \mathcal{S}$.
We also have

$$
\begin{aligned}
V_H^*(s) - \widehat{f}_H(s) &= \langle Q_H^*(s,\cdot) - \widehat{f}_H(s,\cdot), \pi^*(\cdot|s) \rangle_{\mathcal{A}} + \langle \widehat{f}_H(s,\cdot), \pi_H^*(\cdot|s) - \widehat{\pi}_H(\cdot|s) \rangle_{\mathcal{A}} \\
&\leq \langle Q_H^*(s,\cdot) - \widehat{f}_H(s,\cdot), \pi^*(\cdot|s) \rangle_{\mathcal{A}} \\
&= \langle [\mathcal{T}_H V_H^*](s,\cdot) - \widetilde{f}_H(s,\cdot) + b_H(s,a), \pi^*(\cdot|s) \rangle_{\mathcal{A}} \\
&= \langle [\mathcal{T}_H \widehat{f}_{H+1}](s,\cdot) - \widetilde{f}_H(s,\cdot) + b_H(s,a), \pi^*(\cdot|s) \rangle_{\mathcal{A}} \\
&\quad + \langle [\mathcal{T}_H V_{H+1}^*](s,\cdot) - [\mathcal{T}_H \widehat{f}_{H+1}](s,\cdot), \pi_H^*(\cdot|s) \rangle_{\mathcal{A}} \\
&\leq 2\langle b_H(s,\cdot), \pi_H^*(\cdot|s) \rangle_{\mathcal{A}} + \epsilon \\
&\leq \widetilde{O}\left( \frac{\sqrt{\log \mathcal{N}} H^2}{\sqrt{K\kappa}} \right),
\end{aligned}
$$

where the first inequality holds due to the policy $\widehat{\pi}_H$ takes the action which maximizes $\widehat{f}_H$. The second inequality holds due to the Bellman completeness assumption. The last inequality holds due to the property of the bonus function

$$
b_H(z) \leq C \cdot \left( D_{\mathcal{F}_H}(z; \mathcal{D}_H; \widehat{\sigma}_H) \cdot \sqrt{(\beta_H)^2 + \lambda} + \epsilon \beta_H \right)
$$

and Lemma F.1.
Then we do the induction step. Let $R_h = \widetilde{O}\left( \frac{\sqrt{\log \mathcal{N}} H^2}{\sqrt{K\kappa}} \right) \cdot (H - h + 1)$, $\delta_h = (H - h + 1)\delta/(4H^2)$.
We define another event $\mathcal{E}_h^{\mathrm{ind}}$ for induction.

$$
\mathcal{E}_h^{\mathrm{ind}} = \{0 \leq V_h^*(s) - \widehat{f}_h(s) \leq R_h, \forall s \in \mathcal{S}\}.
$$

The above analysis shows that $\mathcal{E}_H \subseteq \mathcal{E}_H^{\mathrm{ind}}$ and $\mathbb{P}[\mathcal{E}_H] \geq 1 - 2\delta_H$. Moreover, $\mathbb{P}[\mathcal{E}_H^{\mathrm{ind}}] \geq 1 - 2\delta_H$
We conduct the induction in the following way. At stage $h$, if $\mathbb{P}[\mathcal{E}_{h+1}] \geq 1 - 2\delta_{h+1}$ and $\mathbb{P}[\mathcal{E}_{h+1}^{\mathrm{ind}}] \geq 1 - 2\delta_{h+1}$, we prove that $\mathbb{P}[\mathcal{E}_h] \geq 1 - 2\delta_h$ and $\mathbb{P}[\mathcal{E}_h^{\mathrm{ind}}] \geq 1 - 2\delta_h$.
Suppose at stage $h$, $\mathbb{P}[\mathcal{E}_{h+1}] \geq 1 - 2\delta_{h+1}$ and $\mathbb{P}[\mathcal{E}_{h+1}^{\mathrm{ind}}] \geq 1 - 2\delta_{h+1}$. We first use Lemma G.1. Let $f = \widetilde{f}_h \in \mathcal{F}_h$, $f' = \bar{f}_h \in \mathcal{F}_h$. We define

$$
\eta_h^k := V_{h+1}^*(s_{h+1}^k) - [\mathbb{P}_h V_{h+1}^*](z_h^k)
$$

$$
D_h^k[f, f'] := 2 \frac{\eta_h^k}{(\widehat{\sigma}_h(z_h^k))^2} \left( f(z_h^k) - f'(z_h^k) \right).
$$

After taking a union bound, we have with probability at least $1 - \delta/(4H^2)$, the following inequality holds,

$$
\sum_{k\in[K]} 2 \frac{\eta_h^k}{(\widehat{\sigma}_h(z_h^k))^2} \left( \widetilde{f}(z_h^k) - \bar{f}(z_h^k) \right) \leq \frac{4}{3} v(\delta)\sqrt{\lambda} + \sqrt{2} v(\delta) + 30 v^2(\delta)
$$

$$+ \frac{\sum_{k\in[K]} \frac{1}{(\widehat{\sigma}_h(z_h^k))^2} \left(\widetilde{f}(z_h^k) - \bar{f}(z_h^k)\right)^2}{4}. \tag{G.8}$$

Next, we use Lemma G.2 at stage $h$. Let $f = \widetilde{f}_h \in \mathcal{F}_h$, $\widetilde{f} = \bar{f}_h \in \mathcal{F}_h$, $f' = \widehat{f}_{h+1} = \{\widetilde{b}\}_{[0, H-h+1]}$, where $\widetilde{b} = \widehat{f}_h - b_h \in \mathcal{F}_h - \mathcal{W}$. We define:

$$\xi_h^k[f'] := f'(s_{h+1}^k) - V_{h+1}^*(s_{h+1}^k) - \left[\mathbb{P}_h(f' - V_{h+1}^*)\right](z_h^k)$$

$$\Delta_h^k[f, \widetilde{f}, f'] := 2\frac{\xi_h^k[f']}{(\widehat{\sigma}_h(z_h^k))^2} \left(f(z_h^k) - \widetilde{f}(z_h^k)\right),$$

After taking a union bound, we have with probability at least $1 - \delta/(4H^2)$, we have

$$\sum_{k\in[K]} 2\frac{\xi_h^k[\widehat{f}_{h+1}]}{(\widehat{\sigma}_h(z_h^k))^2} \left(\widetilde{f}_h(z_h^k) - \bar{f}_h(z_h^k)\right) \le \left(\frac{4}{3}\iota(\delta)\sqrt{\lambda} + \sqrt{2}\iota(\delta)\right) \|f' - V_{h+1}^*\|_\infty^2$$

$$+ \frac{2}{3}\iota^2(\delta)/\sqrt{\log \mathcal{N}_b} + 30\iota^2(\delta)\|\widehat{f}_{h+1} - V_{h+1}^*\|_\infty^2 + \frac{\sum_{k\in[K]} \frac{1}{(\widehat{\sigma}_h(z_h^k))^2}(\widetilde{f}_h(z_h^k) - \bar{f}_h(z_h^k))^2}{4}. \tag{G.9}$$

Let $U_h$ be the event that (G.8) and (G.9) holds simultaneously. On the event $U_h \cap \mathcal{E}_{h+1}^{\mathrm{ind}}$, which satisfies $\mathbb{P}[U_h \cap \mathcal{E}_{h+1}^{\mathrm{ind}}] \ge 1 - 2\delta_{h+1} - 2\delta/H^2 = 1 - 2\delta_h$, we have

$$2\sum_{k\in[K]} \frac{1}{(\widehat{\sigma}_h(z_h^k))^2} \left(r_h^k + \widehat{f}_{h+1}(s_{h+1}^k) - \bar{f}_h(z_h^k)\right) \left(\widetilde{f}_h(z_h^k) - \bar{f}_h(z_h^k)\right)$$

$$= 2\sum_{k\in[K]} \frac{1}{(\widehat{\sigma}_h(z_h^k))^2} \left(r_h^k + \widehat{f}_{h+1}(s_{h+1}^k) - [\mathcal{T}_h\widehat{f}_{h+1}](z_h^k)\right) \left(\widetilde{f}_h(z_h^k) - \bar{f}_h(z_h^k)\right)$$

$$+ 2\sum_{k\in[K]} \frac{1}{(\widehat{\sigma}_h(z_h^k))^2} \left([\mathcal{T}_h\widehat{f}_{h+1}](z_h^k) - \bar{f}_h(z_h^k)\right) \left(\widetilde{f}_h(z_h^k) - \bar{f}_h(z_h^k)\right)$$

$$\le 2\sum_{k\in[K]} \frac{1}{(\widehat{\sigma}_h(z_h^k))^2} \left(r_h^k + \widehat{f}_{h+1}(s_{h+1}^k) - [\mathcal{T}_h\widehat{f}_{h+1}](z_h^k)\right) \left(\widetilde{f}_h(z_h^k) - \bar{f}_h(z_h^k)\right) + 4KL\epsilon$$

$$\le \frac{4}{3}v(\delta)\sqrt{\lambda} + \sqrt{2}v(\delta) + 30v^2(\delta) + \left(\frac{4}{3}\iota(\delta)\sqrt{\lambda} + \sqrt{2}\iota(\delta)\right) \|\widehat{f}_{h+1} - V_{h+1}^*\|_\infty^2$$

$$+ \frac{2}{3}\iota^2(\delta)/\log \mathcal{N}_b + 30\iota^2(\delta)\|\widehat{f}_{h+1} - V_{h+1}^*\|_\infty^2 + 8KL\epsilon + \frac{\sum_{k\in[K]} \frac{1}{(\widehat{\sigma}_h(z_h^k))^2} \left(\bar{f}_h(z_h^k) - \widetilde{f}_h(z_h^k)\right)^2}{2}$$

$$\le \frac{(\beta_h)^2}{2} + \frac{\sum_{k\in[K]} \frac{1}{(\widehat{\sigma}(z_h^k))^2} \left(\bar{f}_h(z_h^k) - \widetilde{f}_h(z_h^k)\right)^2}{2}, \tag{G.10}$$

where the first inequality holds due to the completeness assumption. The second inequality holds due to (G.8) and (G.9). The last inequality holds due to the event $\mathcal{E}_{h+1}^{\mathrm{ind}}$ and the choice of $K \ge \widetilde{\Omega}\left(\frac{\iota(\delta)^2 H^6}{\kappa}\right)$.

However, on the event of $\mathcal{E}_h^c$, we have

$$2\sum_{k\in[K]} \frac{1}{(\widehat{\sigma}(z_h^k))^2} \left(r_h^k + \widehat{f}_{h+1}(s_{h+1}^k) - \bar{f}_h(z_h^k)\right) \left(\widetilde{f}_h(z_h^k) - \bar{f}_h(z_h^k)\right)$$

$$\ge \sum_{k\in[K]} \frac{1}{(\widehat{\sigma}(z_h^k))^2} \left(\bar{f}_h(z_h^k) - \widetilde{f}_h(z_h^k)\right)^2$$

$$> \frac{(\beta_h)^2}{2} + \frac{\sum_{k\in[K]} \frac{1}{(\widehat{\sigma}(z_h^k))^2} \left(\bar{f}_h(z_h^k) - \widetilde{f}_h(z_h^k)\right)^2}{2}.$$

where the first inequality holds due to (G.4). The second inequality holds due to event $\mathcal{E}_h^c$. However, this contradicts with (G.10). We conclude that $U_h \cap \mathcal{E}_{h+1}^{\mathrm{ind}} \subseteq \mathcal{E}_h$, thus $\mathbb{P}[\mathcal{E}_h] \ge 1 - 2\delta_h$.

Next we prove $\mathbb{P}[\mathcal{E}_h^{\mathrm{ind}}] \geq 1 - 2\delta_h$. Suppose the event $U_h \cap \mathcal{E}_{h+1}^{\mathrm{ind}}$ holds, the above conclusion shows that

$$\sum_{k \in [K]} \frac{1}{(\widehat{\sigma}_h(z_h^k))^2} \left( \bar{f}_h(z_h^k) - \widetilde{f}_h(z_h^k) \right)^2 > (\beta_h)^2.$$

We can prove the following result.

$$
\begin{aligned}
Q_h^*(s, a) &= [\mathcal{T}_h V_{h+1}^*](s, a) \\
&\geq [\mathcal{T}_h \widehat{f}_{h+1}](s, a) \\
&\geq \widetilde{f}_h(s, a) - |[\mathcal{T}_h \widehat{f}_{h+1}](s, a) - \widetilde{f}_h(s, a)| \\
&\geq \widetilde{f}_h(s, a) - (\epsilon + |\bar{f}_h(s, a) - \widetilde{f}_h(s, a)|) \\
&\geq \widetilde{f}_h(s, a) - b_h(s, a) \\
&= \widehat{f}_h(s, a),
\end{aligned}
$$

where the first inequality hold due to the event $\mathcal{E}_h^{\mathrm{ind}}$. The second inequality holds due to the triangle inequality. The third inequality holds due to the completeness assumption. The last inequality holds due to the property of the bonus function and

$$\sum_{k \in [K]} \frac{1}{(\widehat{\sigma}_h(z_h^k))^2} \left( \bar{f}_h(z_h^k) - \widetilde{f}_h(z_h^k) \right)^2 \leq (\beta_h)^2.$$

Therefore, $V_h^*(s) \geq \widehat{f}_h(s)$ for all $s \in \mathcal{S}$.
We also have

$$
\begin{aligned}
V_h^*(s) - \widehat{f}_h(s) &= \langle Q_h^*(s, \cdot) - \widehat{f}_h(s, \cdot), \pi^*(\cdot|s) \rangle_{\mathcal{A}} + \langle \widehat{f}_h(s, \cdot), \pi_h^*(\cdot|s) - \widehat{\pi}_h(\cdot|s) \rangle_{\mathcal{A}} \\
&\leq \langle Q_h^*(s, \cdot) - \widehat{f}_h(s, \cdot), \pi^*(\cdot|s) \rangle_{\mathcal{A}} \\
&= \langle [\mathcal{T}_h V_h^*](s, \cdot) - \widetilde{f}_h(s, \cdot) + b_h(s, a), \pi^*(\cdot|s) \rangle_{\mathcal{A}} \\
&= \langle [\mathcal{T}_h \widehat{f}_{h+1}](s, \cdot) - \widetilde{f}_h(s, \cdot) + b_h(s, a), \pi^*(\cdot|s) \rangle_{\mathcal{A}} \\
&\quad + \langle [\mathcal{T}_h V_{h+1}^*](s, \cdot) - [\mathcal{T}_h \widehat{f}_{h+1}](s, \cdot), \pi_h^*(\cdot|s) \rangle_{\mathcal{A}} \\
&\leq 2 \langle b_h(s, \cdot), \pi_h^*(\cdot, s) \rangle_{\mathcal{A}} + \epsilon + R_{h+1} \\
&\leq \widetilde{O} \left( \frac{\sqrt{\log \mathcal{N}} H^2}{\sqrt{K \kappa}} \right) \cdot (H - h + 1) = R_h.
\end{aligned}
$$

where the first inequality holds due to the policy $\widehat{\pi}_h$ takes the action which maximizes $\widehat{f}_h$. The second inequality holds due to the Bellman completeness assumption. The second inequality holds due to the property of the bonus function

$$b_h(z) \leq C \cdot \left( D_{\mathcal{F}_h}(z; \mathcal{D}_h; \widehat{\sigma}_h^2) \cdot \sqrt{(\beta_h)^2 + \lambda} + \epsilon \beta_h \right)$$

and Lemma F.1. The last inequality holds due to the induction assumption. Therefore, we have $U_h \cap \mathcal{E}_{h+1}^{\mathrm{ind}} \subseteq \mathcal{E}_h^{\mathrm{ind}}$ and $\mathbb{P}[\mathcal{E}_h^{\mathrm{ind}}] \geq 1 - 2\delta_h$. Thus we complete the proof of induction.
Finally, taking the union bound of all the $\mathcal{E}_h$, we get the result that with probability at least $1 - \delta/2$, the event $\cup_{h=1}^H \mathcal{E}_h$ holds, i.e for any $h \in [H]$ simultaneously, we have

$$\sum_{k \in [K]} \frac{1}{(\widehat{\sigma}_h(z_h^k))^2} \left( \bar{f}_h(z_h^k) - \widetilde{f}_h(z_h^k) \right)^2 \leq (\beta_h)^2.$$

Therefore, we complete the proof of Lemma F.2. $\qquad\square$

# H  PROOF OF LEMMAS IN SECTION E AND G

## H.1  PROOF OF LEMMA E.1
*Proof of Lemma E.1.* We use Lemma I.1, with the following conditions:

$\bar{D}_h^k[f, f']$ is adapted to the filtration $\bar{\mathcal{H}}_h^k$ and $\mathbb{E}\left[ \bar{D}_h^k[f, f'] \mid \bar{\mathcal{H}}_h^{k-1} \right] = 0$.

$$\left| \bar{D}_h^k[f, f'] \right| \le 2 \left| \bar{\eta}_h^k \right| \max_z |f(z) - [\mathcal{T}_h f'](z)| \le 4H^2 = M.$$

$$\sum_{k \in [K]} \mathbb{E}\left[ \left( \bar{D}_h^k[f, f'] \right)^2 \Big| \bar{z}_h^k \right] = \sum_{k \in [K]} \mathbb{E}\left[ 4 \left( \bar{\eta}_h^k[f'] \right)^2 \Big| \bar{z}_h^k \right] \left( f(\bar{z}_h^k) - [\mathcal{T}_h f'](\bar{z}_h^k) \right)^2 \le (4HK)^2 = V^2.$$

On the other hand,

$$\sum_{k \in [K]} \mathbb{E}\left[ \left( \bar{D}_h^k[f, f'] \right)^2 \Big| \bar{z}_h^k \right] = \sum_{k \in [K]} \mathbb{E}\left[ 4 \left( \bar{\eta}_h^k[f'] \right)^2 \Big| \bar{z}_h^k \right] \left( f(\bar{z}_h^k) - [\mathcal{T}_h f'](\bar{z}_h^k) \right)^2$$
$$\le 8H^2 \sum_{k \in [K]} \left( f(\bar{z}_h^k) - [\mathcal{T}_h f'](\bar{z}_h^k) \right)^2.$$

Then using Lemma I.1 with $v = 1$, $m = 1$, with at least $1 - \delta/(4H^2 \mathcal{N}^2 \mathcal{N}_b^2)$

$$\sum_{k \in [K]} 2\bar{\eta}_h^k[f'] \left( f(\bar{z}_h^k) - [\mathcal{T}_h f'](\bar{z}_h^k) \right) \le i(\delta) \sqrt{2(2 \cdot 8H^2) \sum_{k \in [K]} \left( f(\bar{z}_h^k) - [\mathcal{T}_h f'](\bar{z}_h^k) \right)^2}$$
$$+ \frac{2}{3} i^2(\delta) + \frac{4}{3} i^2(\delta) \cdot 4H^2$$
$$\le (24H^2 + 5) i^2(\delta) + \frac{\sum_{k \in [K]} \left( f(\bar{z}_h^k) - [\mathcal{T}_h f'](\bar{z}_h^k) \right)^2}{2}.$$

We complete the proof of Lemma E.1. $\qquad \square$

### H.2   PROOF OF LEMMA E.2

*Proof of Lemma E.2.* We use Lemma I.1, with the following conditions:

$\bar{D}_h^k[f, f']$ is adapted to the filtration $\bar{\mathcal{H}}_h^k$ and $\mathbb{E}\left[ \bar{D}_h^k[f, f'] \mid \bar{\mathcal{H}}_h^{k-1} \right] = 0$.

$\left| \bar{D}_h^k[f, f'] \right| \le 2 |\bar{\eta}_h^k| \max_z |f(z) - [\mathcal{T}_{2,h} f'](z)| \le 4L^2 = M.$

$\sum_{k \in [K]} \mathbb{E}\left[ \left( \bar{D}_h^k[f, f'] \right)^2 \Big| \bar{z}_h^k \right] = \sum_{k \in [K]} \mathbb{E}\left[ 4(\bar{\eta}_h^k[f'])^2 | \bar{z}_h^k \right] \left( f(\bar{z}_h^k) - [\mathcal{T}_{2,h} f'](\bar{z}_h^k) \right)^2 \le (4L^2 K)^2 = V^2.$

On the other hand,

$$\sum_{k \in [K]} \mathbb{E}\left[ \left( \bar{D}_h^k[f, f'] \right)^2 \Big| \bar{z}_h^k \right] = \sum_{k \in [K]} \mathbb{E}\left[ 4 \left( \bar{\eta}_h^k[f'] \right)^2 \Big| \bar{z}_h^k \right] \left( f(\bar{z}_h^k) - [\mathcal{T}_h f'](\bar{z}_h^k) \right)^2$$
$$\le 8L^4 \sum_{k \in [K]} \left( f(\bar{z}_h^k) - [\mathcal{T}_{2,h} f'](\bar{z}_h^k) \right)^2.$$

Then using Lemma I.1 with $v = 1$, $m = 1$, we have:

$$\sum_{k \in [K]} 2\bar{\eta}_h^k[f'] \left( f(\bar{z}_h^k) - [\mathcal{T}_{2,h} f'](\bar{z}_h^k) \right) \le i'(\delta) \sqrt{2(2 \cdot 8L^4) \sum_{k \in [K]} \left( f(\bar{z}_h^k) - [\mathcal{T}_{2,h} f'](\bar{z}_h^k) \right)^2}$$
$$+ \frac{2}{3} i'^2(\delta) + \frac{4}{3} i'^2(\delta) \cdot 4L^2$$
$$\le (20L^4 + 5) i'^2(\delta) + \frac{\sum_{k \in [K]} \left( f(\bar{z}_h^k) - [\mathcal{T}_h f'](\bar{z}_h^k) \right)^2}{2}$$

We complete the proof of Lemma E.2. $\qquad \square$

### H.3   PROOF OF LEMMA G.1

*Proof of Lemma G.1.* We use Lemma I.1, with the following conditions:

$$D_h^k[f, f'] \text{ is adapted to the filtration } \mathcal{H}_h^k \text{ and } \mathbb{E}\left[ D_h^k[f, f'] \mid \mathcal{H}_h^{k-1} \right] = 0.$$
$$\left| D_h^k[f, f'] \right| \le 2 \left| \eta_h^k \right| \max_z |f(z) - f'(z)| \le 8LH = M.$$
$$\sum_{k \in [K]} \mathbb{E}\left[ \left( D_h^k[f, f'] \right)^2 \Big| z_h^k \right] = 4 \sum_{k \in [K]} \frac{\mathbb{E}\left[ (\eta_h^k)^2 | z_h^k \right]}{(\widehat{\sigma}_h(z_h^k))^4} \left( f(z_h^k) - f'(z_h^k) \right).$$

On the other hand,

$$\sum_{k\in[K]} \mathbb{E}\left[\left(D_h^k[f,f']\right)^2 \Big| z_h^k\right] = 4\sum_{k\in[K]} \frac{\mathbb{E}\left[(\eta_h^k)^2 | z_h^k\right]}{(\widehat{\sigma}_h(z_h^k))^4}\left(f(z_h^k) - f'(z_h^k)\right)^2$$

$$\leq 8\sum_{k\in[K]} \frac{1}{(\widehat{\sigma}_h(z_h^k))^2}\left(f(z_h^k) - f'(z_h^k)\right)^2,$$

where the last inequality holds because of the inequality in Lemma D.4:

$$\mathbb{E}\left[(\eta_h^k)^2 | z_h^k\right] = [\text{Var}_h V_{h+1}^*](s_h^k, a_h^k)$$

$$\leq [\mathbb{V}_h V_{h+1}^*](s_h^k, a_h^k)$$

$$\leq \left(\widehat{\sigma}_h(z_h^k)\right)^2 + \widetilde{O}\left(\frac{\sqrt{\log(\mathcal{N}\cdot\mathcal{N}_b)}H^3}{\sqrt{K\kappa}}\right)$$

$$\leq 2\left(\widehat{\sigma}_h(z_h^k)\right)^2,$$

where we use the requirement that $K \geq \widetilde{\Omega}\left(\frac{\log(\mathcal{N}\cdot\mathcal{N}_b)H^6}{\kappa}\right)$.

Moreover, for any $k \in [K]$,

$$\left|D_h^k[f,f']\right| \leq 2\left|\frac{\eta_h^k}{(\widehat{\sigma}_h(z_h^k))^2}\right|\left|f(z_h^k) - f'(z_h^k)\right|$$

$$\leq 4H\sqrt{D_{\mathcal{F}_h}^2(z_h^k, \mathcal{D}_h, \widehat{\sigma}_h^2)\left(\sum_{k\in[K]} \frac{1}{(\widehat{\sigma}_h(z_h^k))^2}\left(f(z_h^k) - f'(z_h^k)\right)^2 + \lambda\right)}$$

$$\leq \widetilde{O}\left(\frac{4H^2}{\sqrt{K\kappa}}\right)\sqrt{\sum_{k\in[K]} \frac{1}{(\widehat{\sigma}_h(z_h^k))^2}\left(f(z_h^k) - f'(z_h^k)\right)^2 + \lambda}$$

$$\leq \frac{1}{v(\delta)}\sqrt{\sum_{k\in[K]} \frac{1}{(\widehat{\sigma}_h(z_h^k))^2}\left(f(z_h^k) - f'(z_h^k)\right)^2 + \lambda}.$$

The second inequality holds because of the definition of $D^2$ divergence (Definition 3.4). The third inequality holds due to Lemma F.1. The last inequality holds because of the choice of $K \geq \widetilde{\Omega}\left(\frac{v^2(\delta)H^4}{\kappa}\right)$.

Then using Lemma I.1 with $v = 1$, $m = 1$, we have

$$\sum_{k\in[K]} 2\frac{\eta_h^k}{(\widehat{\sigma}_h(z_h^k))^2}\left(f(z_h^k) - f'(z_h^k)\right) \leq v(\delta)\sqrt{16\sum_{k\in[K]} \frac{1}{(\widehat{\sigma}_h(z_h^k))^2}\left(f(z_h^k) - f'(z_h^k)\right)^2 + 2} + \frac{2}{3}v^2(\delta)$$

$$+ \frac{4}{3}v(\delta)\sqrt{\sum_{k\in[K]} \frac{1}{(\widehat{\sigma}_h(z_h^k))^2}\left(f(z_h^k) - f'(z_h^k)\right)^2 + \lambda}$$

$$\leq \frac{4}{3}v(\delta)\sqrt{\lambda} + \sqrt{2}v(\delta) + 30v^2(\delta)$$

$$+ \frac{\sum_{k\in[K]} \frac{1}{(\widehat{\sigma}_h(z_h^k))^2}\left(f(z_h^k) - f'(z_h^k)\right)^2}{4}.$$

We complete the proof of Lemma G.1. $\qquad\qquad\square$

## H.4 PROOF OF LEMMA G.2

*Proof of Lemma G.2.* $\Delta_h^k[f, \widetilde{f}, f']$ is adapted to the filtration $\mathcal{H}_h^k$ and $\mathbb{E}\left[\Delta_h^k[f, \widetilde{f}, f'] \mid \mathcal{H}_h^{k-1}\right] = 0$. We also have

$$\sum_{k\in[K]} \mathbb{E}\left[(\Delta_h^k[f, \widetilde{f}, f'])^2 \Big| z_h^k\right] = 4\sum_{k\in[K]} \frac{\mathbb{E}\left[(\xi_h^k[f'])^2 | z_h^k\right]}{(\widehat{\sigma}_h(z_h^k))^4}\left(f(z_h^k) - f'(z_h^k)\right)^2$$

$$\leq 8 \sum_{k \in [K]} \frac{\|f' - V_{h+1}^*\|_\infty^2}{(\widehat{\sigma}_h(z_h^k))^2} \left( f(z_h^k) - f'(z_h^k) \right)^2.$$

Moreover, for any $k \in [K]$,

$$\left| \Delta_h^k[f, \widetilde{f}, f'] \right| \leq 2 \left| \frac{\xi_h^k[f']}{(\widehat{\sigma}_h(z_h^k))^2} \right| \left| f(z_h^k) - f'(z_h^k) \right|$$

$$\leq 4 \|f' - V_{h+1}^*\|_\infty \sqrt{D_{\mathcal{F}_h}^2(z_h^k, \mathcal{D}_h, \widehat{\sigma}_h^2) \left( \sum_{k \in [K]} \frac{1}{(\widehat{\sigma}_h(z_h^k))^2} \left( f(z_h^k) - f'(z_h^k) \right)^2 + \lambda \right)}$$

$$\leq \widetilde{O}\left( \frac{H}{\sqrt{K\kappa}} \right) \cdot \|f' - V_{h+1}^*\|_\infty \sqrt{\sum_{k \in [K]} \frac{1}{(\widehat{\sigma}_h(z_h^k))^2} \left( f(z_h^k) - f'(z_h^k) \right)^2 + \lambda}$$

$$\leq \frac{\|f' - V_{h+1}^*\|_\infty}{\iota(\delta)} \sqrt{\sum_{k \in [K]} \frac{1}{(\widehat{\sigma}_h(z_h^k))^2} \left( f(z_h^k) - f'(z_h^k) \right)^2 + \lambda}$$

The second inequality holds because of the definition of $D^2$ divergence (Definition 3.4). The third inequality holds due to Lemma F.1. The last inequality holds because of the choice of $K \geq \widetilde{\Omega}\left( \frac{\iota^2(\delta)H^4}{\kappa} \right)$.

Then using Lemma I.1 with $v = 1$, $m = 1/\log \mathcal{N}_b$, with probability at least $1 - \delta/(4H^2\mathcal{N}^3\mathcal{N}_b)$, we have

$$\sum_{k \in [K]} 2 \frac{\xi_h^k[f']}{(\widehat{\sigma}_h(z_h^k))^2} \left( f(z_h^k) - \widetilde{f}(z_h^k) \right) \leq \iota(\delta) \sqrt{8 \sum_{k \in [K]} \frac{\|f' - V_{h+1}^*\|_\infty^2}{(\widehat{\sigma}_h(z_h^k))^2} \left( f(z_h^k) - f'(z_h^k) \right)^2 + 2}$$

$$+ \frac{2}{3}\iota^2(\delta)/\log \mathcal{N}_b + \frac{4}{3}\iota(\delta)\|f' - V_{h+1}^*\|_\infty \sqrt{\sum_{k \in [K]} \frac{1}{(\widehat{\sigma}_h(z_h^k))^2} (f(z_h^k) - f'(z_h^k))^2 + \lambda}$$

$$\leq \left( \frac{4}{3}\iota(\delta)\sqrt{\lambda} + \sqrt{2}\iota(\delta) \right) \|f' - V_{h+1}^*\|_\infty^2 + \frac{2}{3}\iota^2(\delta)/\log \mathcal{N}_b + 30\iota^2(\delta)\|f' - V_{h+1}^*\|_\infty^2$$

$$+ \frac{\sum_{k \in [K]} \frac{1}{(\widehat{\sigma}_h(z_h^k))^2} \left( f(z_h^k) - f'(z_h^k) \right)^2}{4}.$$

We complete the proof of Lemma G.2. $\qquad\qquad\qquad\qquad\qquad\qquad\qquad\qquad\qquad\qquad\square$

# I AUXILIARY LEMMAS

**Lemma I.1** (Agarwal et al. 2023). Let $M > 0$, $V > v > 0$ be constants, and $\{x_i\}_{i \in [t]}$ be a stochastic process adapted to a filtration $\{\mathcal{H}_i\}_{i \in [t]}$. Suppose $\mathbb{E}[x_i | \mathcal{H}_{i-1}] = 0$, $|x_i| \leq M$ and $\sum_{i \in [t]} \mathbb{E}[x_i^2 | \mathcal{H}_{i-1}] \leq V^2$ almost surely. Then for any $\delta, \epsilon > 0$, let $\iota = \sqrt{\log \frac{(2\log(V/v) + 2) \cdot (\log(M/m) + 2)}{\delta}}$, we have

$$\mathbb{P}\left( \sum_{i \in [t]} x_i > \iota \sqrt{2\left( 2\sum_{i \in [t]} \mathbb{E}[x_i^2 | \mathcal{H}_{i-1}] + v^2 \right)} + \frac{2}{3}\iota^2 \left( 2\max_{i \in [t]} |x_i| + m \right) \right) \leq \delta.$$

**Lemma I.2** (Regret Decomposition Property, Jin et al. 2021b). Suppose the following inequality holds,

$$\left| [\mathcal{T}_h \widehat{f}_{h+1}](z) - \widetilde{f}_h(z) \right| \leq b_h(z), \forall z = (s, a) \in \mathcal{S} \times \mathcal{A}, \forall h \in [H],$$

the regret of Algorithm 1 can be bounded as

$$V_1^*(s) - V_1^{\widehat{\pi}}(s) \leq 2\sum_{h=1}^{H} \mathbb{E}_{\pi^*}\left[ b_h(s_h, a_h) \mid s_1 = s \right].$$

Here $\mathbb{E}_{\pi^*}$ is with respect to the trajectory induced by $\pi^*$ in the underlying MDP.

**Lemma I.3** (Azuma-Hoeffding inequality, Cesa-Bianchi & Lugosi 2006). Let $\{x_i\}_{i=1}^n$ be a martingale difference sequence with respect to a filtration $\{\mathcal{G}_i\}$ satisfying $|x_i| \leq M$ for some constant $M$, $x_i$ is $\mathcal{G}_{i+1}$-measurable, $\mathbb{E}[x_i|\mathcal{G}_i] = 0$. Then for any $0 < \delta < 1$, with probability at least $1 - \delta$, we have

$$\sum_{i=1}^n x_i \leq M\sqrt{2n\log(1/\delta)}.$$

