# OpenReview forum: "Pessimistic Nonlinear Least-Squares Value Iteration for Offline Reinforcement Learning"
_ICLR.cc/2024/Conference — ICLR 2024 poster_

### Official Review · Reviewer_38or · 2023-10-30

**Soundness:** 2 fair
**Presentation:** 3 good
**Contribution:** 2 fair
**Rating:** 5
**Confidence:** 3

**Summary:**

This paper considers offline RL with non-linear function approximation. The authors propose a new pessimism-based algorithm, Pessimistic Nonlinear Least-Square Value Iteration, to solve this problem. The proposed algorithm is oracle-efficient, and achieves instance-dependent regret bound characterized by the newly-proposed $D^2$ divergence.

**Strengths:**

- The paper proposes an algorithm for offline RL with non-linear function approximation that has instance-dependent regret guarantees, which is new in literature.
- The author extend techniques used for linear MDPs (including variance-weighted ridge regression and reference-advantage decomposition) to nonlinear function classes.
- The paper is generally well-written and clear.

**Weaknesses:**

- This paper poses a very strong converage assumption (Assumption 3.5), which may not be realistic in practice. In contrast, most pessimism-based offline RL papers that I'm aware of adopt weaker partial coverage assumptions. I wonder how valuable it is to prove an instance-dependent bound when we have to impose a uniform coverage condition.
- The lower bound cited by the paper only works for linear cases. The authors do not provide a matching lower bound to showcase the optimality of the proposed algorithm with general function approximations.
- The authors do not provide any numerical experiments in the paper. In my opinion, adding numerical experiments can better showcase the benefits of the proposed algorithm compared to other offline RL algorithms with general function approximation.

**Questions:**

- Which part of the proof requires uniform data coverage while partial data coverage does not work?
- This paper considers general function approximation, yet I'm unclear what types of function classes & instance structures will satisfy the conditions listed in the paper. Can you give some concrete examples beyond linear MDPs?
- Can you provide a lower bound for general function approximations?

---

> ### Author Response · Authors · 2023-11-20
> **Response to Reviewer 38or**
>
> Thank you for your valuable feedback. We address your questions point-by-point.
>
> **Q1**: How valuable is it to prove an instance-dependent bound when we have to impose a uniform coverage assumption? Which part of the proof requires uniform data coverage while partial data coverage does not work?
>
> **A1**: In this work, we aim to achieve an optimal dependency on the complexity of the function class. For this purpose, we utilize the technique of reference-advantage decomposition to overcome the challenge posed by the additional error from uniform convergence over the function class $\mathcal F_h$. However, the uniform coverage assumption is essential in the reference-advantage decomposition when we prove the advantage part is dominated by the reference part. In detail, in the proof of Lemma F.1 in Appendix G.1, we need the uniform coverage condition to bound the last inequality of Eq. (G.2). Note that Lemma F.1 provides an upper bound on the weighted $D^2$-divergence, and plays a pivotal role in proving Lemma F.2 through reference-advantage decomposition. The uniform coverage assumption is also required in existing works on the optimal complexity for offline RL with linear function approximation [1][2]. How to relax the uniform coverage assumption to partial coverage assumption while still achieving the optimal statistical efficiency remains an open problem and we will study it in the future.
>
> ----
> **Q2**: Can you provide a lower bound for general function approximations?
>
> **A2**: That’s a good question. Currently we don’t know how to prove a lower bound for general function approximation in our framework. In most studies of RL with general function approximation, they do not provide a lower bound dependent on the complexity of the function class. As far as we know, the only exception is DEC [3], which is a framework for online RL and their application to offline RL remains open. How to obtain a lower bound for general function approximations is an interesting future direction, especially for offline RL.
>
> ----
> **Q3**: The authors do not provide any numerical experiments in the paper. In my opinion, adding numerical experiments can better showcase the benefits of the proposed algorithm compared to other offline RL algorithms with general function approximation.
>
> **A3**: The main focus and contribution of our work is on the theoretical side of offline RL. We plan to implement the proposed offline RL algorithm using general function approximation and compare it with existing empirically strong baselines on offline RL benchmarks in our future work.
>
> ----
> **Q4**: More concrete examples beyond linear MDPs.
>
> **A4**: Besides the linear MDP, the differentiable function class studied in [4] is also an example covered by our general function class. We have added a comment in Remark 3.7 in the revision.
>
> ----
> [1] Xiong et al. 2023 Nearly minimax optimal offline reinforcement learning with linear function approximation: Single-agent MDP and markov game, ICLR.
>
> [2] Yin et a;. 2022a Near-optimal Offline Reinforcement Learning with Linear Representation: Leveraging Variance Information with Pessimism. ICLR
>
> [3] Foster et al. 2021  The statistical complexity of interactive decision making.
>
> [4] Yin et al. 2022,  Offline reinforcement learning with differentiable function approximation is provably efficient. ICLR

---

> ### Author Response · Authors · 2023-11-22
> **Follow up with Reviewer 38or**
>
> Dear Reviewer 38or,
>
> We greatly appreciate your valuable feedback and are reaching out to see whether there have been any remaining questions that you have. If our response and revision have addressed your questions and concerns, we kindly request you to re-evaluate our work. Thank you!

---

> ### Author Response · Authors · 2023-11-23
> **Gentle Reminder**
>
> Dear Reviewer 38or
>
> As the deadline for the author-reviewer discussion is fast approaching, we would like to kindly inquire whether our response and revision have adequately addressed your questions and concerns. We really appreciate your feedback.

---

### Official Review · Reviewer_Fsc5 · 2023-11-01

**Soundness:** 3 good
**Presentation:** 4 excellent
**Contribution:** 3 good
**Rating:** 6
**Confidence:** 3

**Summary:**

This paper proposes an offline RL algorithm called Pessimistic Nonlinear Least-Square Value Iteration (PNLSVI) for general function approximation. It introduces a type of D$^2$-divergence to quantify the uncertainty of the offline dataset, and proves an instance-dependent regret bound that has a tight dependence on the function class complexity (its covering number).

**Strengths:**

+ This work fills a gap of designing efficient offline RL algorithms with general function approximation.
+ It extends the concept of D$^2$-divergence in online RL to offline RL.
+ It generalizes reference-advantage decomposition to general function approximation.

**Weaknesses:**

- The work is overall incremental. The key techniques are largely known, either from online RL or offline RL. I can see that there are technical barriers in directly extending them to the problem of offline RL with general function approximation, but such extensions are mostly not too difficult.
- Improving the regret by a square-root of $d$ is quite standard for reference-advantage decomposition. This is more about the previous paper not doing the best job than developing truly novel method/analysis.

**Questions:**

- How to interpret Thm 5.1?
- When is Thm 5.1 better than instance-independent regret bounds?
- Your coverage assumption is weaker, but does it really matter in practice?
- You assume that dataset is produced by a single BP. In reality, this may not always be true. What is the impact of different BPs?

---

> ### Author Response · Authors · 2023-11-20
> **Response to Reviewer Fsc5**
>
> Thank you for your positive feedback. We address your questions as follows.
>
> **Q1**: The work is overall incremental. The key techniques are largely known, either from online RL or offline RL. I can see that there are technical barriers in directly extending them to the problem of offline RL with general function approximation, but such extensions are mostly not too difficult.
>
> **A1**: We agree with the reviewer that the extension of RL with linear function approximation to general function approximation is in general not too difficult given so many recent developments. However, as acknowledged by the reviewer, there are still numerous technical barriers that require effort, and our work is dedicated to addressing these challenges.
>
> In pursuit of optimal statistical complexity in offline RL, we delve into leveraging variance information within the general function approximation setting—a previously unexplored domain. We believe our contribution in this regard is quite significant.
>
> ----
> **Q2**: Improving the regret by a square-root of $d$ is quite standard for reference-advantage decomposition. This is more about the previous paper not doing the best job than developing truly novel method/analysis.
>
>
> **A2**: We agree with the reviewer that improving the regret by $\sqrt{d}$ is anticipated when doing reference-advantage decomposition. However, such a technique has never been explored in offline RL with general function approximation, and we are the first to extend this decomposition technique to this setting.
>
> ----
> **Q3**: How to interpret Thm 5.1?
>
> **A3**: Thm 5.1 establishes an upper bound for the suboptimality of our policy $\hat \pi$. It is an instance-dependent result because it depends on the expected uncertainty, which is characterized by the weighted $D^2$-divergence along the trajectory. Both the trajectory and the weight function are based on the optimal policy and the optimal value function, respectively. Moreover, our result has an optimal dependency on the complexity of the function class when it is specialized to the linear case, characterized by the cardinality of the function class $\tilde O(\sqrt{\log \mathcal N})$.
>
> ----
> **Q4**: When is Thm 5.1 better than instance-independent regret bounds?
>
> **A4**: The suboptimality bound in Theorem 5.1 is always no worse than the corresponding instance-independent regret bounds under the same setting. In the worst case, $D_{ \mathcal F_h}$ $(z_h,\mathcal D_h, \[\mathbb V_h V^*_{h+1}\](\cdot,\cdot)) $ in Theorem 5.1 can be bounded by $\tilde O(\frac{H}{\sqrt{K\kappa}})$ due to Lemma F.1, which reduces to the instance-independent regret bound.
>
> ----
> **Q5**: Your coverage assumption is weaker, but does it really matter in practice?
>
> **A5**: Our goal in comparing various coverage assumptions is to provide a comprehensive review of the assumptions prevalent in the literature. Additionally, we want to ensure that our coverage assumption is no more stringent than that of related works, ensuring a fair comparison of the efficacy of our method against others. It is worth noting that, in practice, verifying the coverage assumption may be challenging. This is precisely why we opt for the weaker coverage assumption—to maximize its practical satisfaction to the greatest extent.
>
> ----
> **Q6**: You assume that dataset is produced by a single BP. In reality, this may not always be true. What is the impact of different BPs?
>
>
> **A6**: The key part of our proof is Assumption 3.5, where we make assumptions on the distribution of the dataset. For simplicity, we consider the single behavior policy situation, which satisfies Assumption 3.5. But it doesn't really matter as long as we have an assumption similar to Assumption 3.5.

---

> > ### Comment · Reviewer_Fsc5 · 2023-11-22
> >
> > Thank you for the clarification. I'll keep my score.

---

> > > ### Author Response · Authors · 2023-11-22
> > > **Thank you**
> > >
> > > Thank you for your support! We are very happy that our response has clarified your questions.

---

### Official Review · Reviewer_KfKk · 2023-11-01

**Soundness:** 3 good
**Presentation:** 2 fair
**Contribution:** 2 fair
**Rating:** 6
**Confidence:** 4

**Summary:**

This paper studies offline reinforcement learning with non-linear function approximation. It proposes an oracle-efficient algorithm, Pessimistic Nonlinear Least-Square Value Iteration (PNLSVI), for offline RL with non-linear function approximation. The algorithmic design comprises three innovative components: (1) a variance-based weighted regression scheme that can be applied to a wide range of function classes, (2) a subroutine for variance estimation, and (3) a planning phase that utilizes a pessimistic value iteration approach. The algorithm enjoys a regret bound that has a tight dependency on the function class complexity and achieves minimax optimal instance-dependent regret when specialized to linear function approximation.

**Strengths:**

This paper proposes a pessimism-based algorithm Pessimistic Nonlinear Least-Square Value Iteration (PNLSVI) designed for nonlinear function approximation, which strictly generalizes the existing pessimism-based algorithms for both linear and differentiable function approximation (Xiong et al., 2023; Yin et al., 2022b). The algorithm is oracle-efficient, i.e., it is computationally efficient when there exists an efficient regression oracle and bonus oracle for the function class (e.g., generalized linear function class). In addition, this paper introduces a new type of D2-divergence to quantify the uncertainty of an offline dataset, which naturally extends the role of the elliptical norm seen in the linear setting and the D2-divergence.

**Weaknesses:**

1. Even though there is an Appendix C explaining the computational aspect of computing the bonus, it seems only address the first bullet point in 4.3. How to computationally efficiently obtain the condition for the second bullet point in 4.3?

2. This paper claims it generalizes over the differentiable parametric models in [Yin et al. 22b], would the main theorem 5.1 improves the results obtained in  [Yin et al. 22b]?

3. This paper seems to be closely related to [Alekh et al. 23], however is not enough discussion about. May I consider this paper as an offline version of [Alekh et al. 23]? If not, what are differences?

**Questions:**

Please answer the questions above.

---

> ### Author Response · Authors · 2023-11-20
> **Response to Reviewer KfKk**
>
> Thank you for your positive feedback. We address your questions point-by-point.
>
> **Q1**: How to computationally efficiently obtain the condition for the second bullet point in 4.3?
>
> **A1**: In Appendix C, we actually discuss how to efficiently compute the optimization problem $\max\{ |f_1(z_h)-f_2(z_h)|, f_1,f_2 \in \mathcal F_h: \sum_{k \in [K] } \frac{(f_1(z_h^k)- f_2(z_h^k))^2}{(\hat \sigma_h(s_h^k,a_h^k))^2} \leq (\beta_h)^2\}$.  With the definition of the D^2 divergence, for any $f_1,f_2 \in \mathcal F_h $ satisfying $\sum_{k \in [K]}\frac{1}{(\hat\sigma_h(z_h^k))^2}(f_1(z_h^k)-f_2(z_h^k))^2 \le (\beta_h^2)$, we have
>     $|f_1(z_h)-f_2(z_h)| \le D_{\mathcal F_h}(z;\mathcal D_h; \hat\sigma_h^2) \sqrt{(\beta_h)^2 + \lambda}$.
> Therefore, the second bullet is automatically satisfied. We have provided a more detailed explanation in Appendix C in the revision.
>
> ----
> **Q2**: This paper claims it generalizes over the differentiable parametric models in [1]. Would the main theorem 5.1 improve the results obtained in [1]?
>
> **A2**: Yes, our Theorem 5.1 improves the results obtained in [1]. The main improvement is the dependency on the complexity of the function class. Consider the covering number (or the cardinality for finite function class) for the differentiable parametric model studied in [1], which is covered by our framework. It has been proved in [1] that the covering number is $\tilde O(d)$, and their result has a linear dependency on $d$. Using our Theorem 5.1, we get a $\tilde O (\sqrt{d})$ dependency on $d$, which has a $\sqrt {d}$ improvement compared with the results obtained in [1].
>
> ----
>
> **Q3**: This paper seems to be closely related to [Alekh et al. 23], however is not enough discussion about. May I consider this paper as an offline version of [Alekh et al. 23]? If not, what are differences?
>
> **A3**: Alekh et al. 23 [2] is an inspiring work in the realm of online RL with general function approximation. We have drawn considerable insights from their research, particularly in the definition of $D^2$-divergence and the selection of the bonus function, which we have discussed. However, there is a key distinction beyond the apparent difference between the settings of online and offline RLs. Unlike the approach in [2], which involves a very intricate policy derived from the optimistic Q-value function or, under specific conditions, the overly optimistic Q-value function, our algorithm employs a simple greedy policy. This eliminates the need for an overly pessimistic Q-value function (the counterpart of the overly optimistic Q-value function as we’re doing offline RL) in addition to the pessimistic Q-value function. Our approach simplifies the algorithm significantly, distinguishing it from a direct offline version of the online algorithm in [2].
>
> ----
> [1] Yin et al. 2022,  Offline reinforcement learning with differentiable function approximation is provably efficient. ICLR
>
> [2] Agarwal et al. 2023, $VOQL$: Towards optimal regret in model-free rl with nonlinear function approximation. COLT

---

### Official Review · Reviewer_FxDp · 2023-11-05

**Soundness:** 4 excellent
**Presentation:** 3 good
**Contribution:** 3 good
**Rating:** 8
**Confidence:** 3

**Summary:**

This paper considers the problem of offline reinforcement learning in finite-horizon MDPs with general function approximation, from a theoretical perspective.
The main assumptions are:
- Bellman completeness: completeness of the of the value-function hypothesis class under first and second order Bellman optimality operators, possibly with misspecification
- Uniform coverage of the hypothesis class by the offline dataset, in a novel form suited to general nonlinear value function approximation.
- Access to a computational oracle (nonlinear least-squares regression)

The algorithm is UCBVI-like and combines pessimism variance-weighted least squares regression, where in turn the variance is estimated pessimistically from a fold of the dataset. It is oracle-efficient.

The main theoretical result is an upper bound on the simple regret that scales with the square root of the logarithm of the cardinality (or covering number) of the hypothesis class, and with inverse data coverage as measured by a D^2 divergence. The latter makes the bound instance-dependent. In this way, an instance-dependent regret bound is achieved in a more general setting and with weaker assumptions w.r.t. previous works.

**Strengths:**

The paper is well written and clear. Theoretical claims are supported by detailed and well commented proofs, the main technical tools are highlighted and explained, and the assumptions are clear.
The summary of the state of the art is particularly complete and detailed. Both the strengths and the limitations of the work are properly highlighted and discussed.

**Weaknesses:**

I just have some minor remarks:
1. In section 3 I think you went a bit too far with the abuse of notation when defining Bellman operators. Is f a function of the state or the state and action? How can you define the relationship between the Q and the V function of a fixed policy by Bellman's *optimality* operator?
2. Assumption 3.2 is followed/complemented by other assumptions that are just given in-line. I suggest to state them as separate assumptions to improve clarity.
3. You define $\mathcal{N}$ as the cardinality of the hypothesis class but refer to it as the "covering number". The two are not the same, and the only covering argument that I could find was in Remark 5.3. for the linear case.

Typos:
- page 4 "closed to the optimal value function"
- page 6: "construct this variance estimator with..."

**Questions:**

Could you give an intuitive reason for why you also need an *under*estimation of the variance? Also, does it make sense to call it "pessimistic" in this case? A smaller variance estimate has the effect of inflating the value estimate since you are using inverse variance weights, so it would seem more optimistic than pessimistic.

---

> ### Author Response · Authors · 2023-11-20
> **Response to Reviewer FxDp**
>
> Thank you for your strong support. We will address your questions as follows.
>
> **Q1**: In section 3, I think you went a bit too far with the abuse of notation when defining Bellman operators. Is $f$ a function of the state or the state and action? How can you define the relationship between the Q and the V function of a fixed policy by Bellman's optimality operator?
>
> **A1**: Thank you for your suggestion. We have defined $f(s) = \max_{a}f(s,a)$ and added a footnote to clarify the two slightly abused notations $f(s)$ and $f(s,a)$.
>
> For the Bellman operator, we have two separate equations. One is the Bellman equation for value functions of a fixed policy. The other is the Bellman optimality equation for optimal value functions.
>
> ----
> **Q2**: Assumption 3.2 is followed/complemented by other assumptions that are just given in-line. I suggest to state them as separate assumptions to improve clarity.
>
> **A2**: Thanks for your suggestion. We have revised it to be two separate assumptions (i.e., Assumptions 3.2 and 3.3): realizability and completeness.
>
> ----
> **Q3**: You define $\mathcal{N}$ as the cardinality of the hypothesis class but refer to it as the "covering number". The two are not the same, and the only covering argument that I could find was in Remark 5.3. for the linear case.
>
> **A3**: Sorry for the confusion. In this paper, for simplicity, we assume the function class is finite, and we simply use $\mathcal N$ to denote the cardinality of the hypothesis class. For the infinite function class, we can use the covering number to replace the cardinality. Note that the covering number will be reduced to the cardinality when the function class is finite. We have added a clarification to avoid any misunderstanding.
>
> ----
> **Q4**:  Intuitive reason for why you also need an underestimation of the variance. A smaller variance estimate has the effect of inflating the value estimate since you are using inverse variance weights, so it would seem more optimistic than pessimistic.
>
> **A4**: In fact, we only need an accurate enough estimation of the variance. In Eq. (6.1), we prove that the variance estimator is very close to the true variance (same order for large enough $K$). Therefore, we can use it as the weight (more precisely, inverse weight) for weighted regression. We have removed “pessimistic” from “pessimistic variance estimator” to avoid any confusion.
>
> ----
> **Q5**: Typos: page 4 "closed to the optimal value function"
> page 6: "construct this variance estimator with..."
>
>
> **A5**: Thanks for pointing out our typos. We have fixed them.

---

### Meta-Review · Area_Chair_GN3B · 2023-12-07

**Metareview:**

The paper examines the incorporation of pessimism for non-linear function approximation, under an assumption of uniform coverage. It improve upon prior art in terms of dependency on the function complexity and matches the lower bound when applied to the linear setting. The framework allows for instance-dependent results. The result is interesting, although fairly incremental. A lower bound for the general case is not presented,

**Justification For Why Not Higher Score:**

Incremental result

**Justification For Why Not Lower Score:**

The paper makes a meaningful contribution in nonlinear offline RL in terms of theoretical statistical efficiency

---

### Decision · Program_Chairs · 2024-01-16

Accept (poster)